DESY 22-074

# Electromagnetic Couplings of Axions

**Anton V. Sokolov, Andreas Ringwald**

*Deutsches Elektronen-Synchrotron DESY, Notkestr. 85, 22607 Hamburg, Germany*

*E-mail:* anton.sokolov@desy.de, andreas.ringwald@desy.de

ABSTRACT: We show that, contrary to assertions in the literature, the main contribution to the axion-photon coupling need not be quantized in units proportional to $e^2$. In particular, we discuss a loophole in the argument for this quantization and then provide explicit counterexamples. Hence, we construct a generic axion-photon effective Lagrangian and find that the axion-photon coupling may be dominated by previously unknown Wilson coefficients. We show that this result implies a significant modification of conventional axion electrodynamics and sets new targets for axion experiments. At the core of our theoretical analysis lies a critical reexamination of the interactions between axions and magnetic monopoles. We show that, contrary to claims in the literature, magnetic monopoles need not give mass to axions. Moreover, we find that a future detection of an axion or axion-like particle with certain parameters can serve as evidence for the existence of magnetically charged matter.

# 1  Introduction

Axions are very well motivated candidates for physics beyond the Standard model (SM), since they provide an elegant solution to the strong CP problem [1–4] and can naturally explain the abundance and properties of dark matter [5–8]. Even more, particles which have axion-like interactions with photons can explain the astrophysical anomaly of TeV transparency of the Universe associated with the propagation of high energy photons from distant TeV blazars and BL Lac type objects [9–11][1]. The axion-photon couplings indicated by the latter anomaly are however much larger than the ones predicted in the relevant mass region in conventional axion models of KSVZ [13, 14] or DFSZ [15, 16] type. This tension has been reconciled recently by the authors of this article who found that the anomalous TeV transparency of the Universe, as well as the axion hint from the cooling of horizontal branch stars in globular clusters [17], can be accounted for in the general hadronic model for the axion [18, 19]. The axion-photon coupling in the latter model, however, seems to contradict the lore established by previous works which discuss the possible range of axion-photon couplings in minimal models [2] – namely that the main contribution to these couplings must be quantized in some units proportional to the electron charge squared $e^2$ [20, 23, 24]. In this work, we show that there is really no contradiction, by extending the results obtained in Ref. [20] and finding that the main contribution to the axion-photon coupling need not be quantized in the above-mentioned units even in minimal axion models.

Our central result which allows us to go beyond the conventional argument about the quantization of the axion-photon coupling stems from a critical reexamination of the interactions between axions and magnetic monopoles. It has been long believed that these interactions are necessarily induced by the Witten effect [25, 26]. What we show is that there are more possibilities, so that the shift of the axion field need not induce electric charge on monopoles. The corresponding non-conventional electromagnetic couplings of axions enter the axion-photon effective field theory (EFT) whenever one admits the "no global symmetries" [27–31] and "completeness of the charge spectrum" [29, 30, 32] conjectures of quantum gravity, since these conjectures imply the existence of magnetic monopoles with any charge allowed by the Dirac-Schwinger-Zwanziger (DSZ) quantization condition [33–35]. The same new interaction terms enter the low energy EFT of axion-like particles (ALPs) – Nambu-Goldstone bosons of any spontaneously broken global $U(1)$ symmetry. We derive phenomenological consequences of the new electromagnetic couplings of the axion and ALPs, showing that they would represent new, distinctive features, which are possible to detect in various axion experiments. We argue that the detection of axions or ALPs with such features would provide circumstantial experimental evidence for the existence of magnetically charged matter.

The article is structured as follows: in sec. 2, we give a brief introduction to the physics of magnetic monopoles with emphasis on the overwhelming theoretical evidence for

---

[1]Note however that a recent astrophysical study [12] of the polarization of light from magnetic white dwarfs excludes most of the parameter region relevant for the TeV transparency hint.

[2]Minimal models exclude constructions with mixing of multiple axions [20] or clockwork mechanism [21, 22].

their existence; in sec. 3, we review the existing argument about the quantization of the axion-photon coupling, indicate a loophole in it and discuss the general structure of the electromagnetic interactions of axions; in sec. 4, we review a local-Lagrangian QFT with magnetic charges and its classical limit, classify all marginal operators within it, discuss CP-violation and description of the Rubakov-Callan process therein; in sec. 5, we build a generic axion-photon EFT and discuss the structure of different axion-photon couplings; in sec. 6, we discuss the phenomenology of the new axion-photon couplings and their implications for axion search experiments; finally, in sec. 7, we conclude.

## 2  Magnetic and electric charges in quantum theory

In 1894, Pierre Curie noted that the existence of magnetic monopoles would be perfectly consistent with the classical theory of electromagnetism [36]. His motivation for considering magnetic charges stemmed from the desire to give magnetism an analogous status to the one of electricity. Now we know that this desire was never to come true: all magnetic phenomena observed so far can be perfectly described by the motion and interactions of electrically charged particles. However, the fundamental quantum theory which provided for such a successful electric description of the magnetic phenomena has also revolutionized our views on magnetic monopoles. Whereas Pierre Curie and his contemporaries regarded magnetic charges as a purely phenomenological construct, Paul Dirac argued in 1931 that the quantum theoretical formalism itself suggests their existence [33]. In particular, he pointed out that, in a consistent quantum theory, the change in the phase of a wave function around any closed curve must be the same modulo $2\pi n$ for all the wave functions, where $n$ is an integer. In case $n = 0$ one recovers the standard gauge principle, introduced earlier by Hermann Weyl [37]. This is the case where there are only electrically but no magnetically charged particles in the theory and which is known to be realized in the Standard model (SM) of particle physics. The quantum theory itself however does not tell us any reason for why only the case $n = 0$ should be realized in nature, so that for a generic quantum mechanical model of particle physics one would expect that any $n$ is allowed. As Dirac showed, the $n \neq 0$ case corresponds to a model with both electric and magnetic monopoles involved. Moreover, he found that the corresponding electric ($q$) and magnetic ($g$) gauge charges are necessarily related via the quantization condition $qg = 2\pi n$, which conveniently explains quantization of the electric charge observed in nature. Note that this is still the simplest explanation for the latter phenomenon to date, since it follows directly from the formalism of quantum mechanics (QM) given one does not put any additional restrictions on the quantum states of the theory.

One can wonder if the results which Dirac obtained in the framework of QM can be derived in the more fundamental formalism of quantum field theory (QFT). Does a generic QFT of gauge interactions predict quantization of the electric gauge charge? The answer is positive and it was found by Daniel Zwanziger [38, 39]. In order to address this question, Zwanziger had to revisit the very basis of any Poincaré-invariant QFT – irreducible unitary representations of the Poincaré group under which particles of the theory can transform. One-particle irreducible representations were studied in 1939 by Eugene

Wigner [40], however what Zwanziger found is that there exist also two-particle irreducible representations. The latter are parameterized by an angular momentum variable, which is quantized, and correspond to pairs of particles, each pair containing both an electric and a magnetic monopole. The quantized angular momentum variable for a given pair is proportional to the product of the corresponding electric and magnetic charges, hence one automatically recovers charge quantization. Even more, a QFT of gauge interactions which allows for the two-particle irreducible representations was explicitly constructed in the case of the Abelian gauge group, both in Hamiltonian and Lagrangian formulations, in the works by Julian Schwinger and Daniel Zwanziger [41, 42]. The latter authors also showed that in an Abelian gauge theory a particle can have both electric and magnetic charges, i.e. it can be a dyon, in which case the quantization condition is generalized: $q_i g_j - q_j g_i = 2\pi n$ for any pair of particles $(i, j)$ [34, 35]. This means the right statement is not that the known charged particles have no magnetic charge, as it is usually claimed, but rather that their magnetic charges happen to be proportional to their electric charges. One of the essential features of both Schwinger's and Zwanziger's formulations is the introduction of two four-potentials for the description of the photon field, one of which couples to the electric and another to the magnetic currents. The advantage of the Zwanziger formulation is that it is based on a Lagrangian which preserves locality and which treats electric and magnetic variables symmetrically. We will review Zwanziger's Lagrangian formulation and take advantage of it later in this article.

So far we have been discussing generic magnetic monopoles, i.e. essentially the possibility of having two different kinds of gauge charges deeply rooted in the formalism of both QM and QFT. Let us now consider 't Hooft-Polyakov magnetic monopoles – topological solitons arising in the spontaneously broken symmetry phase of purely electric non-Abelian gauge theories, discovered by Gerardus 't Hooft and Alexander Polyakov in 1974 [43, 44]. These topological solitons were called magnetic monopoles because they create a monopole-like magnetic field far from their cores. Note however that they are more complicated constructs compared to their fundamental counterparts which we discussed earlier. It is important that the difference can reveal itself even at energies much lower than the inverse monopole core size, for which one would expect 't Hooft-Polyakov monopoles to behave similarly to fundamental ones due to the identical monopole-like configuration of the long-range magnetic field. The reason for this is that the instanton effects of the full non-Abelian theory are not suppressed on monopoles [45], thus contributing an extra rotor degree of freedom to the EFT describing infrared (IR) physics [46]. The best known phenomenological implication of this extra degree of freedom is the Rubakov-Callan effect [45, 47]: as Valery Rubakov and Curtis Callan showed in the beginning of 1980s, the Grand Unified Theory (GUT) magnetic monopole can induce proton decay at a strong interaction rate. If one is interested only in the processes which do not involve the rotor degree of freedom, the 't Hooft-Polyakov monopoles behave similarly to Dirac monopoles in the IR, i.e. their EFT is given by the above-mentioned Zwanziger theory [48]. Since 't Hooft-Polyakov monopoles (more generally, Julia-Zee dyons [49]) are an inevitable prediction of GUTs, they represent a well motivated case for the existence of magnetic monopoles (more generally, dyons). Explicit constructions show that such dyons can be bosonic as well as fermionic [50–52]. In

this work, we will not adhere to any particular GUT, keeping our discussion as generic as possible. Note that the study of the general properties of magnetic monopoles which hold regardless of their origin is an active area of research, see for instance Refs. [53–59].

The 't Hooft-Polyakov construction is not the only way one can get solitons with magnetic charge in unified theories. A less widely known example, but the one which proves quite illustrative for the discussions in this article, is the Kaluza-Klein (KK) monopole [60, 61]. In this case, the soliton which arises as a solution of the 5-dimensional KK theory [62, 63] reproduces the Dirac magnetic monopole upon reduction to 4 space-time dimensions. An essential distinction from 't Hooft-Polyakov solitons is that there is no dyonic rotor degree of freedom living on KK monopoles, which means that there exists no direct analogue of the Rubakov-Callan effect for them [64].

From what has been already discussed, we see that the existence of magnetically charged matter would fit very well both in the structure of QM, completing the gauge principle, and in the structure of relativistic quantum theory, completing the irreducible unitary representations of the Poincaré group realized in nature. The observed quantization of the electric charge would be explained. Moreover, the existence of magnetic monopoles would be a natural consequence of the unification of fundamental interactions, if the latter unification takes place. This is however not an exhaustive list of arguments in support of the existence of magnetically charged particles. Another strong motivation comes from our understanding of gravity: the consistency of a quantum theory of the latter was shown to imply a number of restrictions on the structure of admissible field theories. In particular, it was argued that there can be no global symmetries in a consistent theory which includes quantum gravity [27–31] and that in such a theory, the charge spectrum is complete [29, 30, 32]. These conjectures were shown to imply the existence of magnetic monopoles with any magnetic charge allowed by the DSZ quantization condition [29, 32].

What are the ways to probe magnetic monopoles experimentally? Many direct search techniques have been proposed [65], such as searches for monopoles bound in matter, searches in cosmic rays, searches at colliders and, in the case of some GUT monopoles, searches via the catalysis of nucleon decay. None of the direct detection experiments have however yielded a conclusive signal so far. Moreover, it is quite difficult to derive accurate exclusion limits on the monopole mass due to the large theoretical uncertainty. In fact, the QFT of magnetic monopoles is an essentially non-perturbative theory and there is still no reliable method to calculate cross-sections of the QFT processes involving magnetic charges. The interpretation of the indirect searches for virtual monopoles at colliders [66] suffers from the same problem. In this work, we will point out a new possible signature for virtual magnetic monopoles, which has the advantage of being independent of any non-rigorous statements within the non-perturbative theory of magnetic charges. In particular, we will show that there is a certain modification of free electrodynamics which, if experimentally detected, would favor the existence of magnetic monopoles. A complication is that such a modification must involve a new hypothetical particle – the axion or, more generally, an ALP.

# 3 Revising the structure of the axion-photon coupling

## 3.1 Previous arguments supporting quantization

The axion is the pseudo Nambu-Goldstone boson of the spontaneously broken $U(1)_{\text{PQ}}$ Peccei-Quinn (PQ) symmetry [1–4]. Since the PQ symmetry is anomalous, the low energy axion Lagrangian generally contains non-derivative couplings of the axion to CP-odd combinations of the gauge fields of the low energy SM:

$$\mathcal{L}_a \supset \frac{1}{4}\, g_{a\gamma\gamma}\, a\, F^{\mu\nu} F^d_{\mu\nu} \;+\; \frac{a g_s^2}{32\pi^2 f_a}\, G^{a\,\mu\nu} G^{d\,a}_{\mu\nu}\,, \tag{3.1}$$

where $a$ is the axion field, $F_{\mu\nu}$ ($G_{\mu\nu}$) is the field strength tensor of the QED (QCD) gauge field, $g_{a\gamma\gamma}$ is the axion-photon coupling, $g_s$ is the coupling constant of QCD, $f_a$ is the axion decay constant; summation over the index $a = 1\dots 8$ for gluons is implied and for any tensor $A_{\mu\nu}$ its Hodge dual is defined as $A^d_{\mu\nu} = \epsilon_{\mu\nu\lambda\rho} A^{\lambda\rho}/2$, where $\epsilon_{0123} = 1$.

Both axion-photon and axion-gluon couplings are probed in various experiments. Interactions of the axion with photons are particularly well constrained. In fact, a large parameter region on the $(g_{a\gamma\gamma}, f_a)$ plane has been excluded already and there are a lot of new different experiments planned which are going to explore the axion-photon coupling further in the nearest future. A natural question then arises: where should we look in the first place? What are the best motivated values for $g_{a\gamma\gamma}$ and $f_a$ from the theoretical viewpoint? In the last years, it has been claimed by many authors that this question can be answered by considering a consistency condition for the axion EFT [20, 23, 24]. The latter condition takes advantage of the fact that the axion is essentially an angular variable with a period $2\pi v_a$, where $v_a$ is the PQ symmetry breaking scale, so that the effective low energy action must be symmetric under the following shifts:

$$a \to a + 2\pi v_a n,\ n \in \mathbb{Z}\,. \tag{3.2}$$

Let us review the argument of Refs. [20, 23]. First, since the topological charge of QCD,

$$Q_t \;=\; \frac{g_s^2}{32\pi^2} \int d^4x\, G^{a\,\mu\nu} G^{d\,a}_{\mu\nu}\,, \tag{3.3}$$

is an integer, symmetry of the axion-gluon interaction under the transformation (3.2) requires

$$f_a = v_a/N_{\text{DW}},\ \ N_{\text{DW}} \in \mathbb{Z}\,, \tag{3.4}$$

in which case under the transformation (3.2), the action changes by $2\pi k$, $k \in \mathbb{Z}$, and the path integral is unchanged. Second, since one cannot generically exclude the presence of magnetic monopoles at high energies, it has been claimed that the Witten effect [25] makes the term

$$\theta_{\text{em}} \cdot \frac{e^2}{16\pi^2} \int d^4x\, F^{\mu\nu} F^d_{\mu\nu}\,, \tag{3.5}$$

which enters the QED action, physically relevant [3]. The parameter $\theta_{\text{em}}$ is cyclic with a period that depends on the global structure of the SM group, see Ref. [67]. For the following argument, it is important that the period of $\theta_{\text{em}}$ is always an integer multiple of $2\pi$. Identification of the two similar structures in Eqs. (3.1) and (3.5) restricts the values of the axion-photon coupling $g_{a\gamma\gamma}$ due to the periodicity of the axion field Eq. (3.2):

$$g_{a\gamma\gamma} = \frac{E}{N} \cdot \frac{e^2}{8\pi^2 f_a}, \tag{3.6}$$

where $N = N_{\text{DW}}/2$, $E \in \mathbb{Z}$, and we used Eq. (3.4) in order to relate $g_{a\gamma\gamma}$ to $f_a$. The authors of Refs. [20, 23] then proceed to argue that any contribution to $g_{a\gamma\gamma}$ that is not quantized, i.e. which does not satisfy Eq. (3.6), must be proportional to the mass of the axion squared and can be significantly larger than the order of magnitude of the quantized contribution $e^2/(8\pi^2 f_a)$ only in non-minimal models which introduce new unnecessary energy scales and/or particles.

Let us highlight the step in the derivation where one identifies the two similar $F^{\mu\nu}F^d_{\mu\nu}$ structures in Eqs. (3.1) and (3.5) in the presence of magnetic monopoles. Physically, it is equivalent to stating that electromagnetic interactions between axions and magnetic monopoles are necessarily induced by the Witten effect. What we found is that the latter statement has actually never been consistently derived; moreover, it does not necessarily hold. Before we explain the loophole that has been overlooked, let us briefly review the Witten effect and its widely accepted low energy description since they are central to the following discussion.

## 3.2 Witten effect and its widely accepted low energy description

The Witten effect is an effect in a theory with spontaneously broken non-Abelian gauge symmetry derived in 1979 by Edward Witten [25]. The latter author showed that if the full non-Abelian $SO(3)$ theory with coupling constant $\bar{g}$ and field strength $G_{\mu\nu}$ has a CP-violating parameter $\theta$ in the Lagrangian:

$$\mathcal{L}_\theta = \frac{\theta \bar{g}^2}{32\pi^2} G^{a\,\mu\nu} G^{d\,a}_{\mu\nu}, \ \theta \neq 2\pi n, \tag{3.7}$$

where $n \in \mathbb{Z}$, then 't Hooft-Polyakov monopoles of the broken phase of such a theory get an additional contribution $\delta q$ to their electric charges $q = m\bar{g} + \delta q$, $m \in \mathbb{Z}$:

$$\delta q = \frac{\theta \bar{g}}{2\pi} k, \ k \in \mathbb{Z}, \tag{3.8}$$

which is not quantized in units of $\bar{g}$, but which is proportional to the CP-violating parameter $\theta$.

---

[3] In this argument, one considers only gauge theories formulated on topologically trivial base manifolds with vanishing boundary conditions for the electric and magnetic fields at infinity, so that the term (3.5) does not contribute to any physical processes in the absence of magnetic monopoles. Under additional assumptions of non-trivial topology or special boundary conditions, which could be physically motivated in certain media, one can obtain a quantization law for the axion-photon coupling, too.

One can wonder if there exists a low energy Lagrangian, which would account for the Witten effect, i.e. which would ensure that any monopole-like magnetic field comes together with the monopole-like electric field of strength corresponding to the non-quantized electric charge $\delta q$ from Eq. (3.8). By tree-level matching to the ultraviolet (UV) Lagrangian Eq. (3.7), one obtains the following IR Lagrangian for the non-Abelian theory in the Higgs phase [4]:

$$\mathcal{L} = -\frac{1}{4}F_{\mu\nu}F^{\mu\nu} + \frac{\theta e^2}{32\pi^2}F^{\mu\nu}F^d_{\mu\nu}, \qquad (3.9)$$

where $F_{\mu\nu} = \partial_\mu A_\nu - \partial_\nu A_\mu$ is the electromagnetic field strength tensor, $A_\nu$ is the electromagnetic four-potential and $e$ is the low energy electric gauge coupling.

It seems to be widely believed that the low energy Lagrangian Eq. (3.9) accounts for the Witten effect by itself, without reference to its UV completion, and thus the Witten effect is a generic feature of any $U(1)$ gauge theory. Indeed, if one imposes

$$\partial_\mu F^{d\,\mu\nu} = j^\nu_m, \qquad (3.10)$$

where $j^\nu_m$ is the current of a magnetic charge $kg$, $k \in \mathbb{Z}$, and $g = 4\pi/e$ is the minimal allowed charge of the $SO(3)$ 't Hooft-Polyakov monopole, then one obtains the following equations of motion corresponding to the Lagrangian (3.9):

$$\partial_\mu F^{\mu\nu} = \frac{\theta e}{2\pi}\, k\, (j^\nu_m/(kg))\,. \qquad (3.11)$$

Comparing the latter equation with the expression (3.8) for the non-quantized contribution to the electric charge of the 't Hooft-Polyakov monopole, one concludes that the low energy Lagrangian (3.9) accounts for the Witten effect. This argument, which was to the best of our knowledge first proposed in Ref. [68], is however flawed, which we proceed to show in the next section.

## 3.3 Loophole in the previous arguments

In this section, we discuss a loophole in the theorem of sec. 3.1 on the quantization of the main contribution to the axion-photon coupling. As we already mentioned in the end of sec. 3.1, a possibility to circumvent the theorem arises due to an implicit assumption of the theorem which states that the electromagnetic interactions between axions and magnetic monopoles are necessarily induced by the Witten effect. Let us now explain why the latter proposition has actually never been consistently derived.

Let us start with an auxiliary argument where we consider the Witten effect alone. Suppose that we know nothing about the high energy non-Abelian theory and want to justify the Witten effect merely by means of the low energy EFT. In this EFT, we introduce the topological term, which coincides with the second term of the low energy Lagrangian

---

[4]Note the difference in the coefficients of the $\theta$-terms in Eqs. (3.5) and (3.9). The reason is that defining the Abelian theory of Eq. (3.9) we made a definite choice for the underlying non-Abelian gauge group $SO(3) = SU(2)/\mathbb{Z}_2$. The spectrum of line operators in the non-Abelian theory with this gauge group fixes the period of $\theta$ to be $4\pi$ [67], so that an extra factor of 2 accumulates in the denominator of the corresponding $\theta$-term.

Eq. (3.9). The coefficient $\theta e^2/(32\pi^2)$ is fixed by topology. Derivation of the Witten effect then proceeds on the lines reviewed in the previous section and is done fully within the realm of classical field theory.

The problem is that the latter derivation is not internally consistent. As the Lagrangian (3.9) by definition describes only low energy physics, the fields $F_{\mu\nu}$ entering the corresponding equations of motion, have to be sufficiently weak. Indeed, at strong fields, one would generally expect additional $F_{\mu\nu}$-dependent terms in the Lagrangian, the exact form of which depends on the UV completion. This means that the IR electromagnetic fields are not defined in the small neighborhoods of charges where these fields would become strong, i.e. the support of $F_{\mu\nu}(x)$ must be restricted to exclude the $\varepsilon$-neighborhoods of the worldlines associated to charged particles[5]. Inside such restricted domain, the electromagnetic field satisfies free Maxwell equations at every point, so that $\partial_\mu F^{d\,\mu\nu} = 0$ and the $\theta$-term in the Lagrangian (3.9) corresponds to a merely surface contribution to the action which depends only on the topology of the base manifold. Thus, no Witten effect arises in this careful consideration where one fully respects the domain of applicability of the IR theory.

The failure of the low energy approximation in the neighborhoods of charges is a feature of the IR theories with both electric and magnetic charges, which normally does not occur in field theories without magnetic monopoles. Indeed, usually one can assume that the charged currents are distributed continuously, in which case there are no regions where the field becomes strong. The crucial feature of the theories with magnetic charges is that they do not allow for the charges to be distributed continuously. One way to see this is that the point-like nature of charges is necessary to ensure that the Dirac strings are unobservable [34, 69–71]. Yet another (fully equivalent) way to understand this feature is to consider the Jacobi identity for gauge covariant derivatives: it turns out that it always fails in an irreparable way in the case of the continuous distribution of charges [72]. Thus, strictly speaking, the classical field theory of both electric and magnetic charges is always inconsistent, since one is not allowed to use continuous classical fields to describe charged particles. As we saw in the previous paragraph, this fact leaves us with the necessity of restricting the support of the IR electromagnetic fields and leads to the conclusion that there is really no Witten effect manifest in the Lagrangian (3.9).

Let us now return to the main argument and consider the axion-monopole interactions. The interactions between axions and magnetic monopoles have never been derived from a high energy theory, such as a GUT, but only from the low energy axion EFT, namely from the first term in the Lagrangian (3.1) – see Ref. [26], which is followed by all other works on the subject. Such derivation suffers from the same problem as the low energy derivation of the Witten effect discussed in the previous paragraphs. Indeed, the $\theta$-term in Eq. (3.9) is simply a special $a = \theta f_a$ case of the axion-dependent term in Eq. (3.1). The fact that $a(x)$ can have a general space-time dependence does not change any of the arguments presented for the case of the Witten effect. Thus, we conclude that the Witten-

---

[5]Such procedure suffices if the magnetic charges are non-dynamical, i.e. have large masses, which is a perfectly valid assumption while deriving the Witten effect. If the charges have low masses, a more elaborate procedure is needed to account for the backreaction of the IR fields on the charges.

effect induced interactions between axions and magnetic monopoles are actually absent from the Lagrangian (3.1). As we will show in the next section, this does not mean that such interactions cannot exist – in fact they do arise in some models of non-Abelian gauge fields interacting with axions, but then these interactions are described by a term in the IR Lagrangian that is different from the $aFF^d$ term of the Lagrangian (3.1).

In closing, we see that there is really no robust theoretical argument showing that the electromagnetic interactions between axions and magnetic monopoles are necessarily induced by the Witten effect. Therefore, the argument for the axion-photon coupling quantization from sec. 3.1 need not always hold. In the next sections, we proceed to infer the real structure of low energy electromagnetic couplings of axions.

### 3.4 Two different axion-photon couplings instead of one

#### 3.4.1 Standard axion-photon coupling

In the previous section, we found that the conventional axion-photon coupling,

$$\mathcal{L}_{a\gamma\gamma} \;=\; \frac{1}{4}\, g_{a\gamma\gamma}\, a\, F^{\mu\nu} F^d_{\mu\nu}\,, \tag{3.12}$$

actually does not account for the Witten-effect induced interactions between axions and magnetic monopoles. In particular, for the particular case $a = \theta f_a = \text{const}$, we found that the conventional $\theta$-term of the IR Lagrangian (3.9) does not reproduce the Witten effect, contrary to assertions in the literature. We will now further support these conclusions by deriving the axion Maxwell equations corresponding to the coupling (3.12) and showing that even in the case of a non-zero magnetic current $j_m$, their form is fully standard, without any Witten-effect induced terms.

As it was explained in the previous section, one has to respect the limits of applicability of the low energy approximation, which means that all the fields entering our equations have to be sufficiently weak. As charges have to be localized as opposed to continuously distributed, the latter requirement entails that the equations of motion be written in their integral form [73]:

$$\begin{cases} \displaystyle\int_{\Sigma} \left( \partial_\mu F^{\mu\nu} - g_{a\gamma\gamma}\, \partial_\mu (a F^{d\,\mu\nu}) - j_e^\nu \right) d\Sigma_\nu = 0\,, \\[2ex] \displaystyle\int_{\Sigma} \left( \partial_\mu F^{d\,\mu\nu} - j_m^\nu \right) d\Sigma_\nu = 0\,, \end{cases} \tag{3.13}$$

where $\Sigma$ is an arbitrary 3-surface in space-time. Indeed, these integral equations do not require the assumptions that the medium is continuous and that the field $F_{\mu\nu}$ is defined at every space-time point. In particular, at low energies for which the theory is formulated, the $\varepsilon$-neighborhoods of charged particle worldlines discussed in the previous section are irresolvable by any experiment and therefore we can assume that $F_{\mu\nu}$ is defined everywhere but on these worldlines themselves. This allows us to find the differential equations for

$F_{\mu\nu}(x)$ which are valid almost everywhere:

$$\begin{cases} \partial_\mu F^{\mu\nu} - g_{a\gamma\gamma}\,\partial_\mu a\,F^{d\,\mu\nu} = 0 \\ \partial_\mu F^{d\,\mu\nu} = 0 \end{cases} \quad , \quad x \in M\backslash\{\mathcal{S}_i\}\,, \tag{3.14}$$

where $M$ is the space-time manifold and $\{\mathcal{S}_i\}$ is the set of all the worldlines of charged particles. On $\{\mathcal{S}_i\}$ the differential form of the Eqs. (3.13) does not exist. We see that the first equation in the system (3.13) can be equivalently rewritten as follows:

$$\int_\Sigma \left(\partial_\mu F^{\mu\nu} - g_{a\gamma\gamma}\,\partial_\mu a\,F^{d\,\mu\nu} - j_e^\nu\right) d\Sigma_\nu = 0\,. \tag{3.15}$$

Let us now decompose the Eqs. (3.13) into the zeroth- and first-order equations with respect to the small parameter $\sqrt{s}g_{a\gamma\gamma}$, where $\sqrt{s}$ is the energy scale of our experiments. To achieve this, we expand $F^{\mu\nu} = F_0^{\mu\nu} + F_a^{\mu\nu}$, where $F_0^{\mu\nu} = O(1)$ and $F_a^{\mu\nu} = O(\sqrt{s}g_{a\gamma\gamma})$. The zeroth-order equations are the standard integral Maxwell equations for the field $F_0^{\mu\nu}$ in the presence of magnetic monopoles. It is crucial that at all points where the solution $F_0^{\mu\nu}$ is defined, the differential identity $\partial_\mu F_0^{d\,\mu\nu} = 0$ holds. Using the latter identity, we obtain the following system of first-order $O(\sqrt{s}g_{a\gamma\gamma})$ equations:

$$\begin{cases} \int_\Sigma \left(\partial_\mu F_a^{\mu\nu} - g_{a\gamma\gamma}\,\partial_\mu a\,F_0^{d\,\mu\nu}\right) d\Sigma_\nu = 0\,, \\ \int_\Sigma \partial_\mu F_a^{d\,\mu\nu} d\Sigma_\nu = 0\,. \end{cases} \tag{3.16}$$

Everywhere but on the worldlines of charged particles $\{\mathcal{S}_i\}$, one can then write the corresponding axion Maxwell differential equations for the axion-induced field $F_a^{\mu\nu}(x)$:

$$\begin{cases} \partial_\mu F_a^{\mu\nu} - g_{a\gamma\gamma}\,\partial_\mu a\,F_0^{d\,\mu\nu} = 0 \\ \partial_\mu F_a^{d\,\mu\nu} = 0 \end{cases} \quad , \quad x \in M\backslash\{\mathcal{S}_i\}\,. \tag{3.17}$$

Note that although we had a non-zero magnetic current in the system, there is no Witten-effect induced interaction in the Eqs. (3.16) and (3.17). One can say that the axion field $a$ induces an effective electric four-current in the system $j_{eff}^\nu = g_{a\gamma\gamma}\,\partial_\mu a\,F_0^{d\,\mu\nu}$, however no *real* electric charge is generated by the axion. The effective four-current is obviously conserved $\partial_\mu j_{eff}^\mu = 0$.

### 3.4.2 Witten-effect induced coupling

In the previous sections, we showed that the Witten effect and the Witten-effect induced axion interactions are actually absent from the IR Lagrangians (3.9) and (3.12), respectively. Nevertheless, it is clear that at least in the case of the Witten effect, this effect does occur in the Higgs phase of some non-Abelian models, as demonstrated initially by Witten and

reviewed in sec. 3.2. It is straightforward to write a proper IR Lagrangian describing the Witten effect:

$$\mathcal{L}_{\text{W.e.}} = \left( \bar{j}_e^\mu + \frac{e\theta}{2\pi} \left( j_m^\mu / g \right) \right) A_\mu \,, \tag{3.18}$$

where $\bar{j}_e^\mu$ is the part of the electric current that is quantized in the units of $e$. Indeed, this Lagrangian simply tells us that magnetic monopoles carrying charges $k_i g$ acquire electric charges proportional to $\theta$, with the correct coefficients $ek_i/(2\pi)$. Note that the Lagrangian (3.18) is invariant with respect to the discrete shifts of $\theta$ by $2\pi n$, $n \in \mathbb{Z}$, as it should be since $\theta$ is intrinsically an angular variable. The invariance is achieved due to the quantized part of the electric current $\bar{j}_e^\mu$, the charge of which can be shifted by $ne$ to leave the total electric current unchanged.

Let us now show that the Witten-effect induced interactions of axions arise in the spontaneously broken symmetry phase of the non-Abelian model of sec. 3.2 with the UV Lagrangian (3.7) where the constant $\theta$ is substituted by the dynamical field $a(x)/f_a$. For this, we will generalize the derivation of the Witten effect given in Ref. [25] to the case where one has a dynamical field $a(x)/f_a$ instead of $\theta$. The full Lagrangian for the considered model is:

$$\mathcal{L} = -\frac{1}{4} G^{a\,\mu\nu} G_{\mu\nu}^a + \frac{ae^2}{32\pi^2 f_a} G^{a\,\mu\nu} G_{\mu\nu}^{d\,a} + \frac{1}{2} (D_\mu \phi)^a (D^\mu \phi)^a - \frac{\lambda}{4} \left( \phi^a \phi^a - v^2 \right)^2 \,, \tag{3.19}$$

where $\phi$ is a real scalar field transforming in the adjoint representation of the non-Abelian gauge group, and $D_\mu$ is the gauge covariant derivative. The vacuum expectation value of the field $\phi$ breaks the non-Abelian gauge symmetry spontaneously down to the remaining $U(1)$ subgroup. Assuming for simplicity that a given 't Hooft-Polyakov monopole is fixed at the origin of the coordinates, the scalar field far from it is given by $\phi^a = n^a v$, where $\mathbf{n}$ is a unit radius vector. The generator of the gauge transformations under the linearly realized $U(1)$ gauge subgroup is an infinitesimal isorotation $N$ around $n^a$, which corresponds to the change in the gauge four-potential $\delta A_\mu = (D_\mu \phi)/v$. The $2\pi$ isorotation leaves the system invariant, that is why $N$ has integer eigenvalues. Using the Noether's theorem, one obtains the following expression for $N$:

$$N = \frac{\delta L}{\delta \partial_0 A_\mu^a} \delta A_\mu^a + \frac{\delta L}{\delta \partial_0 \phi^a} \delta \phi^a = \frac{1}{v} \int d^3x \, (D_i \phi)^a G_{0i}^a -$$

$$\frac{e^2}{8\pi^2 f_a v} \int d^3x \, a \, (D_i \phi)^a G_{0i}^{d\,a} = \frac{1}{e} \int d^3x \, \partial_i E_i - \frac{1}{v} \int d^3x \, \phi^a D_i G_{0i}^a -$$

$$\frac{e}{8\pi^2 f_a} \int d^3x \, a \, \partial_i H_i \,, \tag{3.20}$$

where we introduced the standard notations for the $U(1)$ electric and magnetic fields $E_i \equiv e\phi^a G_{0i}^a / v$ and $H_i \equiv e\phi^a G_{0i}^{d\,a} / v$, respectively, and used the Bianchi identity $D_\mu G_{\mu\nu}^{d\,a} = 0$. Next, we transform the second term in the resulting expression by using the equation of motion $D_i G_{0i}^a - e^2 \, \partial_i a \, G_{0i}^{d\,a} / (8\pi^2 f_a) = 0$, and obtain:

$$eN = \int d^3x \left( \partial_i E_i - \frac{e^2}{8\pi^2 f_a} H_i \partial_i a \right) - \frac{e^2}{8\pi^2 f_a} \int d^3x \, a \, \partial_i H_i \,. \tag{3.21}$$

Comparing the first integral with Eq. (3.15) in the case of a purely spatial 3-surface $\Sigma$, we see that this integral equals the total electric charge $Q$. Moreover, using the second equation of the system (3.13), we relate the second integral in the expression for $N$ to the magnetic charge $M$ of the monopole. The result is:

$$eN \;=\; Q \;-\; \frac{e^2}{8\pi^2 f_a}\, a(t,\vec{0}) M \,, \qquad (3.22)$$

from which we infer the allowed electric charge of the monopole:

$$q = ne \;+\; \frac{e\, a(t,\vec{0})}{2\pi f_a}\, k \,, \quad k, n \in \mathbb{Z} \,, \qquad (3.23)$$

where $k$ is the magnetic charge of the monopole in units of $g = 4\pi/e$.

We thus arrived at the intriguing conclusion that, in varying axion backgrounds, the electric charge of the 't Hooft-Polyakov monopole needs not be conserved. The fact that the corresponding IR theory is still fully consistent and gauge-invariant is remarkable and derives from the presence of the so-called dyon collective coordinate – an extra rotor degree of freedom living on the monopole worldline and associated to the instanton effects of the full non-Abelian theory. We will show exactly how this degree of freedom influences the IR physics by writing the gauge-invariant IR Lagrangian for the Witten-effect induced axion interactions in sec. 5.2. We will also show that the latter degree of freedom plays an important role even in the case of constant $\theta$: as we will explain in sec. 4.5, the IR Lagrangian (3.18) can be extended to account for the Rubakov-Callan effect, this extension being fully consistent with the one required to consistently introduce the Witten-effect induced axion interactions.

Although the electric charge of the 't Hooft-Polyakov monopole can vary and the continuity property of the electric current can be violated $\partial_\mu j_e^\mu \neq 0$, the total electric charge of the system is conserved, as it should be due to Lorentz invariance [74]. The mechanism for this conservation was described in Ref. [75] and has to do with the anomaly inflow into the axion topological defects: whilst the electric charge of the monopole changes by $\delta q$, the electric charge on these defects changes by $-\delta q$, in full agreement with the general considerations about the conservation of the total charge of the system. Note however that there is no real electric current between the monopole and the topological defects.

Having obtained and discussed the result (3.23), we can now write the axion Maxwell equations that take into account the Witten-effect induced axion coupling. We denote this coupling as $g_{a\mathrm{Aj}}$ since it is a coupling between an axion, a photon, and a magnetic current. The corresponding generalization of the Eqs. (3.13) is:

$$\begin{cases} \displaystyle\int_\Sigma \left( \partial_\mu F^{\mu\nu} - g_{a\gamma\gamma}\, \partial_\mu a\, F^{d\,\mu\nu} - \bar{j}_e^\nu - g_{a\mathrm{Aj}}\, a\, j_m^{\phi\,\nu} \right) d\Sigma_\nu = 0 \,, \\[2mm] \displaystyle\int_\Sigma \left( \partial_\mu F^{d\,\mu\nu} - j_m^\nu \right) d\Sigma_\nu = 0 \,, \end{cases} \qquad (3.24)$$

where we used the form (3.15) for the first equation; we also denoted the current of the 't Hooft-Polyakov monopoles by $j_m^\phi$, which is a part of the total current $j_m$ of all the

magnetically charged particles. Note that the Witten-effect induced interactions allow axions to generate real electric charge, while on the contrary, the standard axion-photon interactions discussed in the previous section and given by $g_{a\gamma\gamma} \neq 0$, generate only effective electric charge and current. Also note that the coupling $g_{aAj}$ must be quantized in units proportional to $e^2$, in full agreement with the results reviewed in sec. 3.1.

Let us end this section with an explicit example illustrating that the $g_{aAj}$ and $g_{a\gamma\gamma}$ are two different couplings. In particular, consider a KK monopole. As we mentioned previously in sec. 2, a KK monopole does not have an extra dyonic rotor degree of freedom which could give it electric charge. This happens due to the fact that contrary to the case of 't Hooft-Polyakov monopoles, charged fermion fields cannot be excited at the monopole core [64]. This means that there is no direct analogue of the Rubakov-Callan effect for a KK monopole, and also that there can be no Witten-effect induced interactions between axions and KK monopoles. Indeed, the presence of a dyonic rotor degree of freedom is essential for the consistency of the Witten-effect induced interactions [26]; without this extra degree of freedom, these interactions would violate gauge invariance. Therefore, an axion can never interact with a KK monopole via the Witten-effect induced coupling, i.e. $g_{aAj} = 0$ in this case. However, it is also clear that the possibility of the interactions between axions and photons does not depend on whether KK monopoles exist: one could well have $g_{a\gamma\gamma} \neq 0$ in the theory, which would for instance originate from loops of PQ-charged quarks with non-zero electric charges in KSVZ- or DFSZ-like models. Thus, in general, one can have models where $g_{aAj} \neq g_{a\gamma\gamma}$. The KK monopole would then contribute to the current $j_m$, but not to the current $j_m^\phi$, if we follow the notations introduced in Eqs. (3.24).

## 3.5 Even more axion-photon couplings

In the previous sections, we separated the two electromagnetic couplings of axions which have been known before, but which have been considered a single coupling: the standard axion-photon coupling $g_{a\gamma\gamma}$ and the Witten-effect induced coupling $g_{aAj}$. In this section, we finally move to the main result of this article and introduce novel electromagnetic couplings of axions which have never been discussed before.

The main observation which allows us to introduce the new couplings is that the standard axion-photon coupling $g_{a\gamma\gamma}$ given by Eq. (3.12) breaks an important symmetry of the electromagnetic field: the electric-magnetic duality symmetry. This symmetry is an $SO(2)$ rotation in the $(\mathbf{E}, \mathbf{H})$ plane, where $\mathbf{E}$ is the electric field and $\mathbf{H}$ is the magnetic field corresponding to the field strength tensor $F_{\mu\nu}$. It is well-known that the free Maxwell equations are invariant with respect to such rotations. What seems to be less well-known is that the electric-magnetic duality symmetry is a full-fledged global symmetry of the Lagrangian, which has its own conserved Noether charge [76]. Indeed, although the Lagrangian of the electromagnetic field,

$$L_{\mathrm{EM}} \;=\; \int d^3x \,\left( \mathbf{E}^2 - \mathbf{H}^2 \right) , \qquad (3.25)$$

at first sight seems not to be invariant with respect to the aforementioned rotations, it is crucial that the Lagrangian is defined essentially as a functional of the four-potentials $L_{\mathrm{EM}} =$

$L_{\mathrm{EM}}[A_\mu(x)]$, but not of the electric and magnetic fields. The infinitesimal electric-magnetic duality rotation corresponds to the following non-local transformation of the physically significant transverse part of the vector-potential $\mathbf{A}^{\mathrm{T}}$:

$$\delta\mathbf{A}^{\mathrm{T}} = -\theta\,\boldsymbol{\nabla}^{-2}\,\boldsymbol{\nabla}\times\dot{\mathbf{A}}^{\mathrm{T}}\,, \qquad (3.26)$$

which changes the Lagrangian (3.25) by a total time derivative and is thus a symmetry of the theory.

The transformation (3.26) ceases to be a symmetry of the theory in the case where one adds the axion-photon interaction (3.12) to the Lagrangian. For instance, this becomes evident by looking at the axion Maxwell equations (3.17), which feature an effective electric current, but no effective magnetic current. However, at least from the point of view of low energy physics, it is immediately clear that there is no reason why the axion field should break the electric-magnetic duality symmetry in this particular direction, but not in any of the other possible directions. For instance, performing a $\pi/2$ rotation of the direction of the symmetry breaking, we can get the following axion Maxwell equations:

$$\begin{cases} \partial_\mu F_a^{\mu\nu} = 0 \\ \partial_\mu F_a^{d\,\mu\nu} + g_{a\mathrm{BB}}\,\partial_\mu a\,F_0^{\mu\nu} = 0 \end{cases}, \quad x \in M\backslash\{\mathcal{S}_i\}\,, \qquad (3.27)$$

where we denoted the coupling corresponding to this new direction of breaking by $g_{a\mathrm{BB}}$. For example, it is easy to see how this coupling can arise if one formulates the theory of the electromagnetic field in terms of the electric four-potential $B_\mu$, instead of the usual magnetic four-potential $A_\mu$, so that $F_{\mu\nu}^d = \partial_\mu B_\nu - \partial_\nu B_\mu$. In this case, varying the standard axion-photon Lagrangian (3.12) with respect to the dynamical variables of the theory, which are now $B_\mu$, one obtains the axion Maxwell equations of the form (3.27). In such description, the electrically charged particles of the SM couple to the four-potential $B_\mu$, which is now the dynamical variable describing the electromagnetic field, in the same way that the magnetic monopoles couple to the magnetic four-potential $A_\mu$ in the standard formulation of electromagnetism, albeit with a small coupling constant $e$. It is straightforward to incorporate the latter construction into the SM of particle physics, since we are free to choose a suitable description of the gauge theory corresponding to the $U(1)_{\mathrm{EM}}$ electromagnetic subgroup of the full $SU(2)_W \times U(1)_Y$ electroweak group.

By doing the electric-magnetic duality transformation to obtain the new axion Maxwell equations (3.27), we rotated only the standard axion-photon coupling $g_{a\gamma\gamma}$. The Witten-effect induced coupling $g_{a\mathrm{Aj}}$ can also certainly be rotated to obtain a new kind of coupling. Note however that as it was discussed in the previous sections, the Witten-effect induced coupling describes the interactions of axions with photons and 't Hooft-Polyakov magnetic monopoles (or any other magnetically charged objects carrying an extra dyonic rotor degree of freedom). The rotated $g_{a\mathrm{Aj}}$ coupling would describe the interactions of axions with the electric analogues of such magnetic monopoles. In this article, we will not consider such exotic objects and thus we will not discuss the dual analogue of the Witten-effect induced coupling in detail. Also let us stress again that the standard axion-photon coupling $g_{a\gamma\gamma}$ and the Witten-effect induced coupling $g_{a\mathrm{Aj}}$ are two different couplings and the presence of the

coupling $g_{a\mathrm{BB}}$ dual to the coupling $g_{a\gamma\gamma}$ in a given axion model does not entail the presence of the dual Witten-effect induced coupling in this model. The axion-photon interactions described by the system of Eqs. (3.27) certainly do not necessitate the existence of dyonic excitations of electrically charged particles. Not surprisingly, the (non-)existence of such excitations is fully determined only by the properties of the charged particles themselves.

To obtain the system of Eqs. (3.27), we rotated the direction of the electric-magnetic duality symmetry breaking in the system of Eqs. (3.17) by $\pi/2$. One can of course also rotate by any other angle, which gives us the following general form for the axion Maxwell equations:

$$
\begin{cases}
\partial_\mu F_a^{\mu\nu} - g_{a\mathrm{AA}}\, \partial_\mu a\, F_0^{d\,\mu\nu} + g_{a\mathrm{AB}}\, \partial_\mu a\, F_0^{\mu\nu} = 0 \\
\partial_\mu F_a^{d\,\mu\nu} + g_{a\mathrm{BB}}\, \partial_\mu a\, F_0^{\mu\nu} - g_{a\mathrm{AB}}\, \partial_\mu a\, F_0^{d\,\mu\nu} = 0
\end{cases}
, \quad x \in M \backslash \{\mathcal{S}_i\}\,,
\tag{3.28}
$$

where we introduced yet another electromagnetic axion coupling $g_{a\mathrm{AB}}$ and renamed the standard $g_{a\gamma\gamma}$ coupling into $g_{a\mathrm{AA}}$ to conform with the notations for the other couplings.

What are the structure and the possible UV origin of the new couplings? To answer these important questions, we will first consider a suitable formulation of electrodynamics, where the electric-magnetic duality symmetry is implemented in a simple local way contrasted to the non-local implementation (3.26) of the standard approach. Such formulation has already been mentioned in sec. 2 in the context of the quantum relativistic theory of magnetic monopoles – it is the Zwanziger theory [42].

# 4 Quantum electromagnetodynamics

## 4.1 Zwanziger theory

Quantum electromagnetodynamics (QEMD) is the QFT describing interactions of electric charges, magnetic charges and photons. Local-Lagrangian QEMD was constructed by Zwanziger [35]. In the latter theory, the photon is described by two four-potentials $A_\mu$ and $B_\mu$, which are regular everywhere. The gauge group $U(1)$ of electrodynamics is substituted with the new one $U(1)_\mathrm{E} \times U(1)_\mathrm{M}$, where the electric (E) and magnetic (M) factors act in the standard way on $A_\mu$ and $B_\mu$, respectively. One fixes the gauge freedom and restricts the physical states by requiring that they be vacuum states with respect to the free scalar fields[6] $(n{\cdot}A)$ and $(n{\cdot}B)$, where $n^\mu = (0, \vec{n})$ is an arbitrary fixed spatial vector. The right number of degrees of freedom of the photon is preserved due to the special form of the equal-time commutators between the potentials:

$$
[A^\mu(t,\vec{x}), B^\nu(t,\vec{y})] = i\epsilon^{\mu\nu}{}_{\rho 0}\, n^\rho\, (n{\cdot}\partial)^{-1}(\vec{x} - \vec{y})\,,
\tag{4.1}
$$

$$
[A^\mu(t,\vec{x}), A^\nu(t,\vec{y})] = [B^\mu(t,\vec{x}), B^\nu(t,\vec{y})] = -i\,(g_0{}^\mu n^\nu + g_0{}^\nu n^\mu)\,(n{\cdot}\partial)^{-1}(\vec{x} - \vec{y})\,,
\tag{4.2}
$$

where $(n{\cdot}\partial)^{-1}(\vec{x} - \vec{y})$ is the kernel of the integral operator $(n{\cdot}\partial)^{-1}$ satisfying $n{\cdot}\partial\,(n{\cdot}\partial)^{-1}(\vec{x}) = \delta(\vec{x})$:

$$
(n{\cdot}\partial)^{-1}(\vec{x}) = \frac{1}{2} \int_{-\infty}^{\infty} \delta^3\,(\vec{x} - \vec{n}s)\,\varepsilon(s)ds\,,
\tag{4.3}
$$

---

[6]We use the following simplified notations: $a{\cdot}b = a_\mu b^\mu$.

$\varepsilon(s)$ is the signum function. The commutation relations Eqs. (4.1), (4.2) thus make the theory essentially different from the simple case of the gauge theory with two electric $U(1)$ gauge groups, used e.g. in models with a hidden photon. The two four-potentials are not independent and their relation absorbs the non-locality which is inherent to any QFT with both electric and magnetic charges. The Lagrangian of the Zwanziger theory is local and is given by the expression[7]:

$$\mathcal{L} \;=\; \frac{1}{2n^2} \Big\{ [n\cdot(\partial \wedge B)] \cdot [n\cdot(\partial \wedge A)^d] \;-\; [n\cdot(\partial \wedge A)] \cdot [n\cdot(\partial \wedge B)^d] -$$
$$[n\cdot(\partial \wedge A)]^2 \;-\; [n\cdot(\partial \wedge B)]^2 \Big\} \;-\; j_e\cdot A \;-\; j_m\cdot B \;+\; \mathcal{L}_G, \quad (4.4)$$

where $j_e$ and $j_m$ are electric and magnetic currents, respectively, and $\mathcal{L}_G$ is the gauge-fixing part:

$$\mathcal{L}_G \;=\; \frac{1}{2n^2} \Big\{ [\partial\,(n\cdot A)]^2 + [\partial\,(n\cdot B)]^2 \Big\} . \quad (4.5)$$

The Lagrangian (4.4) is invariant under those $SO(2)$ transformations which rotate the two-vectors $(A, B)$ and $(j_e, j_m)$ simultaneously. This symmetry ensures that the absolute directions in the gauge charge space $(q, g)$ are not observable. Note that this is also the symmetry of the DSZ quantization condition which is in fact invariant under a larger $Sp(2) \cong SL(2)$ group of transformations in the gauge charge space. Another important symmetry is however not manifest in the Lagrangian (4.4) – Lorentz invariance seems to be lost. This appearance is in fact deceptive. The reason is intimately connected to the non-perturbativity of the theory and to the DSZ quantization condition. It was shown in Refs. [72, 77] that, after all the quantum corrections are properly accounted for, the dependence on the vector $n_\mu$ in the action $S$ factorizes into a linking number $L_n$, which is an integer, multiplied by the combination of charges entering the quantization condition $q_i g_j - q_j g_i$, which is $2\pi$ times an integer. Since $S$ contributes to the generating functional as $\exp(iS)$, this Lorentz-violating part does not play any role in physical processes. The same result has been obtained directly at the level of amplitudes in the toy model where the magnetic charge is made perturbative [57].

Note that while we introduced the Zwanziger theory using the canonical formalism here, this theory can of course also be formulated using the path integral approach, see Refs. [72, 77–79], where inter alia, the Lorentz invariance and the renormalization of the theory are discussed in detail.

## 4.2 Classical limit and its peculiarities

Let us now show that the classical limit of the theory with the Lagrangian (4.4) indeed corresponds to classical electromagnetism with magnetic currents. The classical equations of motion for the potentials corresponding to the Lagrangian (4.4) are:

$$\frac{n\cdot\partial}{n^2} \left( n\cdot\partial A^\mu \;-\; \partial^\mu n\cdot A \;-\; n^\mu\partial\cdot A \;-\; \epsilon^\mu{}_{\nu\rho\sigma} n^\nu\partial^\rho B^\sigma \right) \;=\; j_e^\mu\,, \quad (4.6)$$

$$\frac{n\cdot\partial}{n^2} \left( n\cdot\partial B^\mu \;-\; \partial^\mu n\cdot B \;-\; n^\mu\partial\cdot B \;-\; \epsilon^\mu{}_{\nu\rho\sigma} n^\nu\partial^\rho A^\sigma \right) \;=\; j_m^\mu\,. \quad (4.7)$$

---

[7]The notations are further simplified: $(a \wedge b)^{\mu\nu} = a^\mu b^\nu - a^\nu b^\mu$, $(a\cdot G)^\nu = a_\mu G^{\mu\nu}$.

They are first-order equations in the time derivative, which allows the two different four-potentials to describe a sole particle – the photon. To transform these equations, it is convenient to use the identity

$$X = \frac{1}{n^2} \left\{ [n \wedge (n \cdot X)] \; - \; [n \wedge (n \cdot X^d)]^d \right\} , \tag{4.8}$$

which holds for any antisymmetric tensor $X$. Namely, assume $X = F$, where $F$ is the field strength tensor introduced such that $n \cdot F = n \cdot (\partial \wedge A)$ and $n \cdot F^d = n \cdot (\partial \wedge B)$. Then, recalling that the scalar expressions $n \cdot A$ and $n \cdot B$ are free fields by definition, one can transform Eqs. (4.6), (4.7) into the Maxwell equations with magnetic currents:

$$\partial_\mu F^{\mu\nu} = j_e^\nu , \tag{4.9}$$

$$\partial_\mu F^{d\,\mu\nu} = j_m^\nu . \tag{4.10}$$

Thus the Lagrangian (4.4) gives us the correct classical equations of motion for the electromagnetic field.

What remains to be seen is whether the classical equations of motion for the charged particles are recovered. Classical expressions for the electric and magnetic currents are:

$$j_e^\nu(x) = \sum_i q_i \int \delta^4(x - x_i(\tau_i))\, dx_i^\nu , \tag{4.11}$$

$$j_m^\nu(x) = \sum_i g_i \int \delta^4(x - x_i(\tau_i))\, dx_i^\nu , \tag{4.12}$$

where $x_i(\tau_i)$ is the trajectory of the i-th particle. Supplementing the Lagrangian (4.4) with the standard kinetic terms for the particles, one obtains the following classical equations of motion for the i-th particle:

$$\frac{d}{d\tau_i} \left( \frac{m_i u_i}{(u_i^2)^{1/2}} \right) = \left( q_i \left[ \partial \wedge A(x_i) \right] + g_i \left[ \partial \wedge B(x_i) \right] \right) \cdot u_i , \tag{4.13}$$

where $u_i^\mu = dx_i^\mu / d\tau_i$. The way the electromagnetic field strength tensor was introduced above ($n \cdot F = n \cdot (\partial \wedge A)$ and $n \cdot F^d = n \cdot (\partial \wedge B)$) and Eqs. (4.9), (4.10) suggest that

$$\partial \wedge A = F + (n \cdot \partial)^{-1} (n \wedge j_m)^d , \tag{4.14}$$

$$\partial \wedge B = F^d - (n \cdot \partial)^{-1} (n \wedge j_e)^d , \tag{4.15}$$

so that the final expression describing the classical force exerted on the i-th particle by the electromagnetic field is:

$$\frac{d}{d\tau_i} \left( \frac{m_i u_i}{(u_i^2)^{1/2}} \right) = \left( q_i F(x_i) + g_i F^d(x_i) \right) \cdot u_i$$
$$- \sum_j (q_i g_j - g_i q_j)\, n \cdot \int (n \cdot \partial)^{-1} (x_i - x_j)\, (u_i \wedge u_j)^d \, d\tau_j . \tag{4.16}$$

This expression correctly accounts for the Lorentz force law only if the non-local term in the second row does not contribute. It is easy to see that the latter term indeed cannot play

any role in classical dynamics, since the support of the kernel $(n\cdot\partial)^{-1}(x_i - x_j)$ is restricted by the condition

$$\vec{x}_i(\tau) - \vec{x}_j(\tau) = \vec{n}s\,, \tag{4.17}$$

which contains three equations, but only two independent variables and is thus satisfied only for exceptional trajectories. At the points of these trajectories where Eq. (4.17) is satisfied, Eq. (4.13) should be solved by continuity, which makes it basically equivalent to the conventional equation for the Lorentz force given by the first row of Eq. (4.16). As it was mentioned before, the full quantum dynamics does not depend on the choice of $\vec{n}$, so that the appearance of the non-local $\vec{n}$-dependent term in Eq. (4.16) is a mere artifact of the classical approximation. In the path integral formulation, exceptional trajectories form a measure zero subset of all trajectories and thus do not contribute to physical amplitudes. Note that this is also true for virtual charged particles, or in other words for intermediate charged particle states in a given amplitude, since the path integral over the corresponding fields can be recast into a path integral over trajectories, see Refs. [72, 80, 81]. The practical prescription which one can use for deriving the classical equations of motion whenever the charged particle, real or virtual, interacts with the electromagnetic field in the initial or final state is simple: in the resulting equations, one should redefine $\partial \wedge A$ and $\partial \wedge B$ by continuity, i.e. substitute $\partial \wedge A \to F$ and $\partial \wedge B \to F^d$.

### 4.3 Marginal operators of QEMD

Let us consider the QEMD Lagrangian (4.4) from the EFT perspective. In particular, we want to find all independent marginal operators respecting the gauge invariance of the theory and preserving the number of degrees of freedom of QEMD. Such operators can be constructed from the gauge-invariant tensors and the vector $n_\mu$. For now, we will not consider the operators containing the gauge currents $j_e$ and $j_m$, which will be discussed in the next section. We find six classes of dimension four operators, each class containing operators of the form $\text{tr}(X\cdot Y)$ and $(n\cdot X)(n\cdot Y)$, where $X$ and $Y$ can stand for any of the two tensors $\partial \wedge A$ and $\partial \wedge B$. From the identity (4.8), one can find the relation between the operators pertaining to the same class:

$$\text{tr}(X\cdot Y) = \frac{2}{n^2}\left[(n\cdot X^d)(n\cdot Y^d) - (n\cdot X)(n\cdot Y)\right]. \tag{4.18}$$

Let us name the classes depending on the pair $(X, Y)$:

$$
\begin{aligned}
x \quad &\text{for} \quad (\partial \wedge A,\, \partial \wedge B)\,, \quad y \quad \text{for} \quad (\partial \wedge A,\, [\partial \wedge B]^d)\,, \\
\alpha \quad &\text{for} \quad (\partial \wedge A,\, \partial \wedge A)\,, \quad \beta \quad \text{for} \quad (\partial \wedge B,\, \partial \wedge B)\,, \\
a \quad &\text{for} \quad (\partial \wedge A,\, [\partial \wedge A]^d)\,, \quad b \quad \text{for} \quad (\partial \wedge B,\, [\partial \wedge B]^d)\,.
\end{aligned}
$$

The members of the same class are distinguished by indices:

$$x_1 \equiv \frac{2}{n^2} \left( n{\cdot}(\partial \wedge A) \right) \left( n{\cdot}(\partial \wedge B) \right) ,$$

$$x_2 \equiv \frac{2}{n^2} \left( n{\cdot}(\partial \wedge A)^d \right) \left( n{\cdot}(\partial \wedge B)^d \right) ,$$

$$x_+ \equiv x_1 + x_2 = \frac{2}{n^2} \left\{ \left( n{\cdot}(\partial \wedge A) \right) \left( n{\cdot}(\partial \wedge B) \right) + \left( n{\cdot}(\partial \wedge A)^d \right) \left( n{\cdot}(\partial \wedge B)^d \right) \right\} ,$$

$$x_- \equiv x_1 - x_2 = -\mathrm{tr} \left( (\partial \wedge A)(\partial \wedge B) \right) ,$$

where we used Eq. (4.18); indices are assigned analogously for the operators in the other five classes. In each of the classes $x$, $y$, $\alpha$ or $\beta$, the basis is formed by any two members. The classes $a$ and $b$ each contain only one operator, since $a_1 = -a_2 = a_-/2$, $a_+ = 0$ and $b_1 = -b_2 = b_-/2$, $b_+ = 0$. Disregarding the source terms, there are thus 10 independent gauge-invariant dimension four operators in the Zwanziger theory, which we choose to be $x_1$, $x_-$, $y_+$, $y_-$, $\alpha_1$, $\alpha_-$, $\beta_1$, $\beta_-$, $a_-$, $b_-$. From these, only three enter the Lagrangian (4.4), the free part of which can be rewritten as follows:

$$\mathcal{L}_\gamma = -\frac{1}{4} \left( y_+ + \alpha_1 + \beta_1 \right) . \tag{4.19}$$

Let us see which operators can be added to this Lagrangian without conflicting with the structure of the theory. The inclusion of the terms $x_- = -\mathrm{tr} \left( (\partial \wedge A)(\partial \wedge B) \right)$, $\alpha_- = -\mathrm{tr} \left( (\partial \wedge A)(\partial \wedge A) \right)$ and $\beta_- = -\mathrm{tr} \left( (\partial \wedge B)(\partial \wedge B) \right)$ is incompatible with the number of degrees of freedom in QEMD, since these operators give rise to second order time derivatives of the four-potentials $A_\mu$ or $B_\mu$ in the classical equations of motion. There is no such problem with the four remaining independent operators, three of which correspond to total derivative terms in the Lagrangian:

$$a_- = -\mathrm{tr} \left\{ (\partial \wedge A)(\partial \wedge A)^d \right\} , \tag{4.20}$$

$$b_- = -\mathrm{tr} \left\{ (\partial \wedge B)(\partial \wedge B)^d \right\} , \tag{4.21}$$

$$y_- = -\mathrm{tr} \left\{ (\partial \wedge A)(\partial \wedge B)^d \right\} , \tag{4.22}$$

and thus do not contribute to the equations of motion. The last operator from our basis is:

$$x_1 = \frac{2}{n^2} \left( n{\cdot}(\partial \wedge A) \right) \left( n{\cdot}(\partial \wedge B) \right) , \tag{4.23}$$

which does modify the equations of motion and should be added to the Zwanziger Lagrangian (4.4) in the EFT approach. Note that since the two four-potentials $A_\mu$ and $B_\mu$ have different parities[8], the operators $a_-$, $b_-$ and $x_1$ are CP-odd, while the operator $y_-$ is CP-even. This means that one can expect the operator $x_1$ to be responsible for CP-violation in QEMD. Let us proceed to the next section to see that $x_1$ is directly related to the Witten effect.

---

[8]Parities of $A_\mu$ and $B_\mu$ can be inferred for instance from Eqs. (4.14) and (4.15).

## 4.4 CP-violation in QEMD

Contrary to QED[9], the theory of QEMD has an intrinsic source of CP-violation. The reason is that the magnetic charge changes its sign under any of the discrete transformations C, P or T [82], so that a dyon with charges $(q, g)$ is mapped into a dyon with charges $(-q, g)$ under a CP-transformation. The spectrum of charges is not CP-invariant if there exists a state $(q, g)$ while its CP-conjugate state $(-q, g)$ is missing. In this case, it is impossible to define a CP transformation in such a way that the theory is invariant under it [58]. Note that due to the DSZ quantization condition,

$$q_i g_j - q_j g_i = 2\pi n, \quad n \in \mathbb{Z}, \tag{4.24}$$

and our choice for the gauge charges carried by the electron $(e, 0)$, any magnetic charge must be quantized in the units of the minimal magnetic charge $g_0 = 2\pi/e$:

$$g_i = n_i^m g_0, \quad n_i^m \in \mathbb{Z}. \tag{4.25}$$

The case of electric charges is however different: what one can infer from the quantization condition (4.24) applied to dyons with charges $(q_1, g_1)$ and $(q_2, g_2)$ is that only the difference of some multiples of the electric charges of dyons is quantized: $n_2^m q_1 - n_1^m q_2 = ne$, $n \in \mathbb{Z}$. The latter condition leads to the quantization of the electric charges themselves only if $q_1 = -q_2$ and $g_1 = g_2$, i.e. only if the theory is CP-invariant. Thus, absolute values of the electric charges introduce a CP-violating parameter $\theta$ into the theory:

$$q_i = \left(n_i^e + \frac{\theta}{2\pi} n_i^m\right) \cdot e, \quad n_i^e \in \mathbb{Z}. \tag{4.26}$$

Since only the total value of the charge, and not any separate contribution, is physical, the parameter $\theta$ introduced in this way is defined on the unit circle $\theta \in [0, 2\pi)$. The additional contribution to the electric charge which is proportional to $\theta$ is in perfect consistency with Eq. (3.8) derived from the Witten effect, which means that in the particular case of 't Hooft-Polyakov monopoles the parameter $\theta$ is the vacuum angle of the full non-Abelian theory.

Let us now find the connection between the CP-violation in QEMD discussed in the previous paragraph and the CP-violating operator $x_1$ introduced in the previous section. We will show that it is possible to remove $\theta$ from the definition of charges (4.26) at the cost of adding the operator $x_1$ with an appropriate coefficient to the kinetic part of the Lagrangian as well as modifying the coefficient in front of the $[n \cdot (\partial \wedge A)]^2$ term. First, we redefine the electric current $j_e \to \bar{j}_e$ so that it contains only the contribution proportional to $n_i^e e$. The QEMD Lagrangian becomes:

$$\mathcal{L} = \frac{1}{2n^2} \left\{ [n \cdot (\partial \wedge B)] \cdot [n \cdot (\partial \wedge A)^d] - [n \cdot (\partial \wedge A)] \cdot [n \cdot (\partial \wedge B)^d] - \right.$$
$$\left. [n \cdot (\partial \wedge A)]^2 - [n \cdot (\partial \wedge B)]^2 \right\} - \left( \bar{j}_e + \frac{e^2 \theta}{4\pi^2} j_m \right) \cdot A - j_m \cdot B. \tag{4.27}$$

---

[9]We assume trivial topology of space-time as there is no evidence to the contrary.

Next, we make the following $SL(2, \mathbb{R})$ transformation in the space of four-potentials:

$$\begin{pmatrix} A \\ B \end{pmatrix} \longrightarrow \begin{pmatrix} 1 & 0 \\ -\dfrac{e^2\theta}{4\pi^2} & 1 \end{pmatrix} \begin{pmatrix} A \\ B \end{pmatrix}. \tag{4.28}$$

The first row of the Lagrangian (4.27), which corresponds to the operator $y_+$ from the previous section, is not affected by this transformation. The second row is transformed yielding the conventional source terms and an extra $x_1$ term as promised:

$$\mathcal{L} = \frac{1}{2n^2} \left\{ [n\cdot(\partial \wedge B)] \cdot [n\cdot(\partial \wedge A)^d] - [n\cdot(\partial \wedge A)] \cdot [n\cdot(\partial \wedge B)^d] - \right.$$

$$\left. \left( 1 + \frac{e^4\theta^2}{16\pi^4} \right) [n\cdot(\partial \wedge A)]^2 - [n\cdot(\partial \wedge B)]^2 + \frac{e^2\theta}{2\pi^2} \left( n\cdot(\partial \wedge A) \right) \left( n\cdot(\partial \wedge B) \right) \right\} -$$

$$\bar{j}_e \cdot A - j_m \cdot B, \tag{4.29}$$

which can be rewritten more compactly in our operator notation:

$$\mathcal{L} = -\frac{1}{4} \left( y_+ + \left( 1 + \frac{e^4\theta^2}{16\pi^4} \right) \alpha_1 + \beta_1 - \frac{e^2\theta}{2\pi^2} x_1 \right) - \bar{j}_e \cdot A - j_m \cdot B. \tag{4.30}$$

Several important comments are in order. First, note that the periodicity of $\theta$ is no longer explicit in the Lagrangian (4.30). In fact, to see the symmetry under $\theta \to \theta + 2\pi$ transformation, we have to account for the implicit dependence of the four-potential $B_\mu$ on $\theta$ arising from the transformation (4.28). The term

$$\frac{1}{4} \cdot \frac{e^2\theta}{2\pi^2} x_1 = \frac{e^2\theta}{4\pi^2 n^2} \left( n\cdot(\partial \wedge A) \right) \left( n\cdot(\partial \wedge B) \right)$$

$$= \frac{e^2\theta}{4\pi^2 n^2} (n\cdot F)(n\cdot F^d) = -\frac{e^2\theta}{16\pi^2} \mathrm{tr}\left( F F^d \right), \tag{4.31}$$

is similar to the conventional QED $\theta$-term (3.5), but is by no means symmetric under the transformation $\theta \to \theta + 2\pi$ by itself. We see that in the theory where magnetic currents are properly included in the Lagrangian of the theory, not only does the term (3.5) lose its total derivative structure, but it is also no longer topological.

The second comment which we would like to make is about Lorentz invariance of QEMD with CP-violation. Although the Lagrangian (4.30) contains an extra term with $n_\mu$-dependence, added to the Zwanziger Lagrangian (4.4), and a change in the coefficient in front of the $\alpha_1$ term, it is clear that the theory is Lorentz-invariant, since one can get rid of the unusual $n_\mu$-dependence by performing a $SL(2, \mathbb{R})$ transformation of the potentials. Since it is always possible to get rid of the $x_1$ term in this way, we see that the three operators $y_+$, $\alpha_1$ and $\beta_1$ entering the Zwanziger Lagrangian (4.19) are indeed the only independent gauge-invariant four-dimensional operators which are relevant for the kinetic part of QEMD. The possible CP-violation is most elegantly accounted for in the expression (4.27) for the QEMD Lagrangian, since in this form the periodicity of the $\theta$-parameter is made obvious. The latter form of the QEMD Lagrangian is also convenient for finding the extension of QEMD which incorporates axions – the endeavor we accomplish in sec. 5.

## 4.5 Rubakov-Callan effect in QEMD

Let us now consider the QEMD of 't Hooft-Polyakov monopoles. As it was already mentioned in sec. 2, the Zwanziger theory provides a good low energy approximation to the dynamics of the 't Hooft-Polyakov monopoles when they are treated as simple point-like magnetic field sources [48]. In the previous section, after introducing the CP-violating parameter $\theta$ into the Zwanziger theory, we identified it with the instanton angle of the UV non-Abelian theory through the Witten effect. Still, the modified Zwanziger Lagrangian (4.27) misses some of the effects associated with the 't Hooft-Polyakov monopoles, since, as we discussed in sec. 2, the latter monopoles cannot be modeled by simple point-like magnetic field sources even in the IR.

Consider instanton effects of the UV non-Abelian theory. At low energies, in the symmetry-broken phase, they are known to be suppressed everywhere, but on 't Hooft-Polyakov monopoles [45]. As a result, some of the good symmetries of the low energy EFT can be violated by unsuppressed instanton-induced effects on the monopole. The most famous example is the Rubakov-Callan effect [45, 47]: the decay of a proton catalyzed by a monopole. A consistent QEMD of 't Hooft-Polyakov monopoles has to account for such instanton effects. To satisfy this requirement, we introduce an extra degree of freedom $\phi(x^\mu)$ into QEMD, which interacts with the electric current $j_e^\mu$ via the following Lagrangian: $\mathcal{L} = (j_e \cdot \partial)\phi$. The field $\phi$ does not contribute to the classical equations of motion, as it should be for a variable describing instanton effects. As the latter effects are localized on the monopole, we require that the interaction Hamiltonian $\mathcal{H} = -(\mathbf{j}_e \cdot \boldsymbol{\nabla})\phi$ vanishes outside the monopole core, so that $\boldsymbol{\nabla}\phi$ is zero everywhere but on the monopole. The latter localization property also means that in our low energy EFT, only s-wave fermions can interact with $\phi$, because wave-functions of scalars and higher partial wave fermions vanish on the monopole due to the centrifugal barrier [83, 84] [10].

Let us now show that the interaction Hamiltonian $\mathcal{H}$ introduced in the previous paragraph can provide a valid description for the Rubakov-Callan effect. For this, we take advantage of the work by Joseph Polchinski [46]. In the latter work, the author showed that the Rubakov-Callan effect can be described as an interaction between s-wave fermions and a rotor coordinate $\alpha(t)$. One can show that this description is equivalent to ours as long as one identifies $\alpha(t)$ with the temporal dependence of $e\phi$. For the sake of comparison, consider the case of a left-handed Weyl fermion $\chi$ interacting with an $SU(2)$ monopole. The only part of the electric current contributing to $\mathcal{H}$ is associated to s-wave fermions $j_e^i = e\,\bar{\chi}_{(s)}^k \sigma^i \chi_{(s)}^k/2$, where $k$ is a flavor index, so that the theory can be reduced to $(1+1)$ dimensions:

$$H = -\int d^3x \, (\mathbf{j}_e \cdot \boldsymbol{\nabla})\,\phi = -\frac{e}{2}\int\limits_0^{+\infty} dr \left(\xi_+^\dagger \xi_+ - \xi_-^\dagger \xi_-\right)\partial_r \phi = \int\limits_{-\infty}^{+\infty} dr \, \psi_k^\dagger \alpha q'(r)\psi_k \,, \quad (4.32)$$

where, following the notations of Ref. [46], we define $\xi_\pm$ spinors as charge eigenstates, $\psi_k(\pm r) \equiv \xi_\pm^{(k)}(r)$; $q(r\,|\,r < -r_0) = 1/2$ and $q(r\,|\,r > r_0) = -1/2$; $r_0$ is the size of the

---

[10]In this section, we assume the magnetic monopole to be a scalar particle, since to our knowledge this is the only case which has been studied in the literature on the Rubakov-Callan effect.

monopole core; we omitted the terms which are suppressed by the high energy scale $1/r_0$. One sees that the interaction Hamiltonian (4.32) is equivalent to the one used in Ref. [46][11]. Thus, to account for the Rubakov-Callan effect, the source term of the QEMD Lagrangian has to be modified as follows: $-j_e \cdot A \rightarrow -j_e \cdot (A - \partial\phi)$. In the case of non-zero $\theta$, one obtains:

$$\mathcal{L} \supset -\left(\bar{j}_e + \frac{e^2\theta}{4\pi^2} j_m\right) \cdot (A - \partial\phi) \,. \tag{4.33}$$

The term $e^2\theta \, (j_m \cdot \partial) \, \phi/4\pi^2$ corresponds to the $\theta$-term in the worldline action for the collective coordinate $e\phi$:

$$S_{[e\phi]} \supset \sum_i \int_{\gamma_i} \frac{\theta}{2\pi} \, d(e\phi) \,, \tag{4.34}$$

where $\gamma_i$ are the monopole worldlines.

There is yet another way to understand why in the case of 't Hooft-Polyakov monopoles, the $\theta$-term of the QEMD Lagrangian (4.27) has to be modified. In particular, consider the case of a varying $\theta$. It is clear that in the full non-Abelian theory the coupling of the new pseudoscalar field $\theta$ to $GG^d$ is legitimate. However, varying $\theta$ in the Lagrangian (4.27) would be inconsistent with electric charge conservation: the gauge invariance of the theory would require $\partial_\mu \theta = 0$. A way to resolve this paradox is to introduce a Stückelberg field $\phi$ localized on the monopole, so that $j_m \cdot (A - \partial\phi)$ is gauge-invariant [85], which leads us again to the coupling (4.33). Note that the dependence of the vacuum energy on $\theta$, which was calculated in Ref. [46] using the low energy theory with the interaction Hamiltonian (4.32), agrees with the high energy non-Abelian theory result obtained in dilute instanton gas approximation $V(\theta) \propto -\cos\theta$. A similar dependence was found in Ref. [86] where the authors took advantage of the worldline action (4.34) and computed the self-energy of $\phi$.

## 5 Generic low energy axion-photon EFT

### 5.1 Anomalous axion-photon interactions

Let us now find the extension of QEMD which incorporates axions. We first limit ourselves to the CP-conserving axion interactions, which means that the dimension five operators containing the axion field are obtained from the CP-odd dimension four operators of QEMD: $a_-$, $b_-$ and $x_1$. Axion EFT must be symmetric under the transformation $a \rightarrow a + 2\pi v_a n$, $n \in \mathbb{Z}$, which suggests that we use the operator $j_m A$ instead of $x_1$, since $ax_1$ would not have the discrete shift symmetry required, as outlined in sec. 4.4. The operator $j_m A$ corresponds to the Witten-effect induced axion interaction and we postpone its discussion until the next section, limiting ourselves to the pure axion-photon couplings first. Thus, the Lagrangian

---

[11]The other two terms in the Hamiltonian of the model of Ref. [46] containing only the collective coordinate $\alpha$ and its canonical momentum $\Pi$ correspond to the potential and kinetic energy terms for $\phi$, respectively.

for a generic CP-conserving axion-photon EFT is[12]:

$$\mathcal{L} = \frac{1}{2n^2} \left\{ [n \cdot (\partial \wedge B)] \cdot [n \cdot (\partial \wedge A)^d] - [n \cdot (\partial \wedge A)] \cdot [n \cdot (\partial \wedge B)^d] - [n \cdot (\partial \wedge A)]^2 - \right.$$
$$\left. [n \cdot (\partial \wedge B)]^2 \right\} - \frac{1}{4} g_{aAA} \, a \, \mathrm{tr} \left\{ (\partial \wedge A)(\partial \wedge A)^d \right\} - \frac{1}{4} g_{aBB} \, a \, \mathrm{tr} \left\{ (\partial \wedge B)(\partial \wedge B)^d \right\} , \quad (5.1)$$

or written in a more compact operator notation:

$$\mathcal{L} = -\frac{1}{4} (y_+ + \alpha_1 + \beta_1) + \frac{1}{4} g_{aAA} \, a \, a_- + \frac{1}{4} g_{aBB} \, a \, b_- . \quad (5.2)$$

The coefficients $g_{aAA}$ and $g_{aBB}$ cannot be determined by symmetry arguments, since both the $a_-$ and $b_-$ terms are total derivatives. In the physically motivated case where our space-time is locally topologically trivial and where electric and magnetic fields vanish sufficiently fast at infinity, the total derivative nature of the $a_-$ and $b_-$ terms ensures axion shift symmetry regardless of their coefficients. Note that in the topologically non-trivial case, or inside a medium with special boundary conditions, one could obtain quantization laws for the coefficients $g_{aAA}$ and $g_{aBB}$ using the discrete shift symmetry requirement. As only topology of the base manifold, but not presence of magnetic currents, contributes to the quantization of these couplings, we see that in the framework of this paper, the effects on the structure of couplings exhibited by non-trivial topology and by magnetic monopoles are separated. Whereas the axion couplings introduced in this section are necessarily quantized only in the case of non-trivial topology, the Witten-effect induced coupling, which we will discuss in the next section, is quantized only due to the presence of a magnetic current.

To compute the coefficients $g_{aAA}$ and $g_{aBB}$ in specific models, we can take advantage of the fact that these terms arise from the anomalous divergence of the Peccei-Quinn current, so that $g_{aAA}$ and $g_{aBB}$ are determined by the $U(1)_{\mathrm{PQ}} (U(1)_{\mathrm{E}})^2$ and $U(1)_{\mathrm{PQ}} (U(1)_{\mathrm{M}})^2$ anomalies, respectively[13]:

$$g_{aAA} = \frac{E e^2}{4\pi^2 v_a} , \quad E = \sum_\psi q_\psi^2 \cdot d(C_\psi) , \quad (5.3)$$

$$g_{aBB} = \frac{M g_0^2}{4\pi^2 v_a} , \quad M = \sum_\psi g_\psi^2 \cdot d(C_\psi) , \quad (5.4)$$

where $E$ and $M$ are electric and magnetic anomaly coefficients, respectively; $q_\psi$ and $g_\psi$ are electric and magnetic charges of heavy PQ-charged fermions $\psi$ in units of $e$ and $g_0$, respectively; $d(C_\psi)$ is the dimension of the color representation of $\psi$. Due to the DSZ quantization condition, $g_0 \gg e$ so that the Wilson coefficient $g_{aBB}$ is expected to dominate the axion-photon coupling.

Let us now consider the CP-violating axion interactions. We have not yet taken advantage of the CP-even four-dimensional operator $y_-$, which can be coupled to the axion since

---

[12]Note that the Lagrangian is essentially a functional of the four-potentials and therefore it is meaningless to rewrite it in terms of electric and magnetic fields.

[13]For the detailed derivation, see Appendix A.

the resulting CP-odd five-dimensional operator $ay_-$ respects the axion shift symmetry. The corresponding term in the Lagrangian is:

$$\mathcal{L}_{\cancel{CP}} \supset -\frac{1}{2} g_{a\mathrm{AB}}\, a\,\mathrm{tr}\left\{(\partial \wedge A)(\partial \wedge B)^d\right\}, \tag{5.5}$$

where the coefficient $g_{a\mathrm{AB}}$ cannot be determined by symmetry arguments, unless one considers non-trivial topology or special boundary conditions in medium, see previous discussion for the case of the $g_{a\mathrm{AA}}$ and $g_{a\mathrm{BB}}$ couplings. In specific models, $g_{a\mathrm{AB}}$ is determined by the $U(1)_{\mathrm{PQ}} U(1)_{\mathrm{E}} U(1)_{\mathrm{M}}$ anomaly. Note that the latter anomaly is non-zero only in the case where the spectrum of dyons violates CP. In this case, the intrinsic CP-violation of high energy QEMD is transferred to the low energy axion-photon EFT after integrating out heavy dyons. As we show in Appendix A, the coefficient $g_{a\mathrm{AB}}$ is given by the following expression:

$$g_{a\mathrm{AB}} = \frac{Deg_0}{4\pi^2 v_a}, \quad D = \sum_\psi q_\psi g_\psi \cdot d(C_\psi), \tag{5.6}$$

where $D$ is the mixed electric-magnetic CP-violating anomaly coefficient, which depends on the spectrum of heavy PQ-charged dyons. The DSZ quantization condition ensures $g_0 \gg e$, so that the CP-violating axion-photon coupling $g_{a\mathrm{AB}}$ is naturally suppressed compared to the CP-conserving $g_{a\mathrm{BB}}$ coupling, but dominates over the CP-conserving $g_{a\mathrm{AA}}$ coupling: $g_{a\mathrm{BB}} \gg |g_{a\mathrm{AB}}| \gg g_{a\mathrm{AA}}$.

Not only do the values of the anomaly coefficients $E$, $M$ and $D$ depend on the details of the UV model, but also the value of the minimal magnetic charge $g_0$ does. While we used $g_0 = 2\pi/e$ for pure QEMD in sec. 4.4, the real low energy theory describing nature involves also the $SU(3)_c$ color gauge group, and the quarks charged under this group have minimal electric charge $|e_0| = e/3$. Naively, this implies that the minimal magnetic charge is $g_0 = 2\pi/|e_0| = 6\pi/e$. However, this is true only if the magnetic monopoles are Abelian, i.e. if they do not carry color magnetic charge. If the monopoles are to the contrary non-Abelian, i.e. if they carry also color magnetic charge[14], the DSZ quantization condition generalizes to include such extra magnetic charges [90–92] and allows for a minimal $U(1)_{\mathrm{M}}$ magnetic charge similar to the one of pure QEMD: $g_0 = 2\pi/e$. In Ref. [18], we built an axion model with heavy PQ-charged fermions $\psi_i$ carrying $SU(3)_{\mathrm{M}}$ color magnetic charges and showed that it indeed solves the strong CP problem. In the explicit calculations of the next sections, we will parameterize the minimal magnetic charge $g_0$ by an integer $\zeta$:

$$g_0 = \frac{2\pi\zeta}{e}, \quad \zeta = \begin{cases} 3, & \psi_i \in U(1)_{\mathrm{E}} \times U(1)_{\mathrm{M}} \times SU(3)_{\mathrm{E}} \\ 1, & \psi_i \in U(1)_{\mathrm{E}} \times U(1)_{\mathrm{M}} \times SU(3)_{\mathrm{M}} \end{cases}. \tag{5.7}$$

As we derived the axion-photon couplings (5.1) and (5.5) from general symmetry arguments, the field $a$ entering our EFT need not be the QCD axion, but could as well correspond to a generic ALP. In this case, Eqs. (5.3), (5.4) and (5.6) need not hold. Nevertheless, the scaling of the corresponding ALP-photon couplings with electric and magnetic elementary

---

[14]Note that contrary to the Abelian case, there are no non-Abelian dyons, i.e. particles which carry both color electric and color magnetic charges [87–89].

charges $e$ and $g_0$ given in Eqs. (5.3), (5.4) and (5.6) persists for any ALP, because with our normalisation of $A_\mu$ and $B_\mu$ four-potentials, the former four-potential always enters the interaction Lagrangian with a factor of $e$ while the latter one always enters the interaction Lagrangian with a factor of $g_0$. This means that for a generic ALP, one still expects the above-mentioned hierarchy of couplings: $g_{a\mathrm{BB}} \gg |g_{a\mathrm{AB}}| \gg g_{a\mathrm{AA}}$.

At the end of this section, let us make an important conceptual remark. Some readers may wonder why a theory where all the magnetic monopoles are very heavy and can be integrated out at low energies, at these energies does not yield a familiar axion-photon EFT Lagrangian given by Eq. (3.12), but rather yields a complicated EFT Lagrangian Eq. (5.1) plus (5.5) that we found in this section. Should not it be that as soon as we move to the effective description, all monopole-induced interactions are absorbed into couplings of IR EFT, which, since there are no magnetic currents anymore, is given by the simple $U(1)$ gauge theory interacting with axions? The answer is no: the Lagrangian (5.1) plus (5.5) is essential in order to account for all possible effects of heavy monopoles. Let us now explain why this is the case. As we mentioned already in sec. 2, any consistent quantum relativistic theory including both electric and magnetic charges is based on two-particle irreducible representations of the Poincaré group, which means that it violates the principle of cluster decomposition. Indeed, no matter how far a given magnetic charge is from a given electric charge, there exists an extra contribution to the angular momentum of the electromagnetic field stemming from their interaction [39, 93]. Since this contribution to the angular momentum of the electromagnetic field does not disappear even when the particles are space-like separated, it is an essentially non-local effect. This means that even if the magnetic monopoles are very heavy and are integrated out to get an effective IR description of the theory, their presence in the UV still affects the electromagnetic field in the IR providing it with an extra angular momentum. In brief, due to the non-local nature of the effect on the electromagnetic field caused by monopoles, this effect does not disappear even if we consider length scales that are much larger than the monopoles' inverse masses. Whenever the axion interacts with the electromagnetic field through loops of heavy magnetic monopoles, the electromagnetic field under consideration has extra angular momentum which leads to a different form of axion-Maxwell equations compared to the case where the axion interacts through loops of heavy electric charges and there is no extra angular momentum in the electromagnetic field.

## 5.2 Witten-effect induced axion interaction

Let us return to the discussion of the CP-conserving $\mathcal{O} = a \, (j_m \cdot A)$ operator of a generic axion EFT. The coefficient in front of this operator is determined by the discrete shift symmetry requirement. The corresponding term in the Lagrangian is obtained by the substitution $\theta \to a/v_a$ in Eq. (4.27):

$$\mathcal{L} \supset - \left( \bar{j}_e + W \cdot \frac{e^2 a}{4\pi^2 v_a} \, j_m \right) \cdot A \,, \tag{5.8}$$

where we also allowed for an arbitrary coefficient $W$. The values of this coefficient are restricted by the discrete shift symmetry of the axion field $a \to a + 2\pi v_a n$, $n \in \mathbb{Z}$. In

particular, the results on the periodicity of $\theta$ obtained in sec. 4.4 show that the axion field has the required discrete shift symmetry iff $(W \cdot n_i^m) \in \mathbb{Z}$ for all $i$, where $n_i^m \in \mathbb{Z}$ is a magnetic charge of the $i$th monopole in units of $g_0$. Therefore, the admissible values of the coefficient $W$ are quantized. Note also that, as we discussed in sec. 4.4 for the analogous case of the $\theta$-parameter, the axion discrete shift symmetry would no longer be explicit if we were to redefine the fields and move the axion dependence into the kinetic part of the Lagrangian.

The term (5.8) is not gauge-invariant unless $\partial_\mu a = 0$, which tells us that our axion EFT has to be modified. A way to restore the gauge invariance is to introduce a Stückelberg field $\phi$ into the Lagrangian:

$$\mathcal{L} \supset - \left( \bar{j}_e + W \cdot \frac{e^2 a}{4\pi^2 v_a} \, j_m \right) \cdot (A - \partial\phi) \, . \tag{5.9}$$

As we discussed in sec. 4.5, such an extra degree of freedom $\phi$ living on the monopole worldline arises naturally while considering the case of 't Hooft-Polyakov monopoles, where it plays the role of the dyon collective coordinate and ensures that the IR theory accounts correctly for the Rubakov-Callan effect. Let us then consider the case where the interaction Lagrangian (5.9) describes the Higgs phase of a non-Abelian gauge theory. Comparing Eq. (5.9) with the $\theta$-term (4.33) of the Higgs phase, we see that the CP-violating $\theta$-parameter of a non-Abelian theory is simply substituted with the axion field $\theta \to a/v_a$, so that the $a\,(j_m \cdot A)$ operator corresponds to the $aGG^d$ operator at high energies. This means that the coupling (5.9) describes Witten-effect induced axion interactions. Indeed, as it was shown in sec. 3.4.2, one obtains the Witten-effect induced coupling $g_{a\mathrm{Aj}}$ between axions and monopoles whenever one considers the spontaneously broken symmetry phase of a non-Abelian theory with $aGG^d$ term.

From the interaction Lagrangian (5.9), we read off the following expression for the Witten-effect induced coupling $g_{a\mathrm{Aj}}$ introduced in sec. 3.4.2:

$$g_{a\mathrm{Aj}} = W \cdot \frac{e^2}{4\pi^2 v_a} \, . \tag{5.10}$$

Due to the quantization property of $W$ discussed previously in this section, we see that the Witten-effect induced coupling is quantized in units proportional to $e^2$. In fact, the structure of the $g_{a\mathrm{Aj}}$ coupling is fully consistent with the well-known result Eq. (3.6) about the quantization of the axion-photon coupling in the presence of monopoles reviewed in sec. 3.1.

Contrary to the three anomalous axion couplings described in the previous section, the coupling (5.9) does not respect the continuous shift symmetry $a \to a + C$, where $C$ is an arbitrary constant. This means that the latter coupling generates a non-flat contribution to the potential for the axion field. Since the axion coupling (5.9) arises in the low energy EFT of a non-Abelian theory with $aGG^d$ interaction, such a contribution to the axion potential is not unexpected: in fact, it has to correspond to the potential created by instantons of the high energy non-Abelian theory. As it was discussed in the end of sec. 4.5, explicit calculations [46, 86] support the latter correspondence. Note, however,

that contrary to the claim made in Ref. [86], additional contribution to the axion potential need not arise in *every* theory of an axion coupled to an Abelian gauge field whenever there are monopoles magnetically charged under the latter field. The axion mass is generated *not* by magnetic monopoles, but always by instantons, even if these instantons happen to live on the monopole worldvolume in the low energy EFT. Indeed, consider the simplest example of a QEMD theory (4.4) which has no extra rotor degrees of freedom. In such a theory, there cannot exist a consistent Witten-effect induced axion coupling, although there can exist the anomalous axion-photon couplings discussed in the previous section. Thus, such a theory has both axions and magnetic monopoles interacting with the Abelian gauge field, but no axion mass is generated through these interactions. A particular example of such kind would be a theory with a KK monopole, see the end of sec. 3.4.2. In general, we see that the anomalous axion-photon couplings and the Witten-effect induced axion coupling are independent. The Witten-effect induced coupling arises only in theories which have instanton degrees of freedom, e.g. in the spontaneously broken symmetry phase of a non-Abelian gauge theory.

## 5.3 Axion Maxwell equations

Having analyzed different axion-photon interactions in the previous two sections, we are now ready to collect them all together in a generic axion-photon EFT Lagrangian:

$$\mathcal{L} = \frac{1}{2n^2} \Big\{ [n\cdot(\partial \wedge B)] \cdot [n\cdot(\partial \wedge A)^d] - [n\cdot(\partial \wedge A)] \cdot [n\cdot(\partial \wedge B)^d] - [n\cdot(\partial \wedge A)]^2 - $$

$$[n\cdot(\partial \wedge B)]^2 \Big\} - \frac{1}{4} g_{aAA} \, a \, \mathrm{tr} \Big\{ (\partial \wedge A)(\partial \wedge A)^d \Big\} - \frac{1}{4} g_{aBB} \, a \, \mathrm{tr} \Big\{ (\partial \wedge B)(\partial \wedge B)^d \Big\} - $$

$$\frac{1}{2} g_{aAB} \, a \, \mathrm{tr} \Big\{ (\partial \wedge A)(\partial \wedge B)^d \Big\} - \Big( \bar{j}_e + g_{aAj} \, a \, j_m^\phi \Big) \cdot (A - \partial \phi) - j_m \cdot B + \mathcal{L}_G \, , \qquad (5.11)$$

or written in a more compact operator notation:

$$\mathcal{L} = -\frac{1}{4} \Big( y_+ + \alpha_1 + \beta_1 - g_{aAA} \, a \, a_- - g_{aBB} \, a \, b_- - 2 \, g_{aAB} \, a \, y_- \Big) - $$
$$\Big( \bar{j}_e + g_{aAj} \, a \, j_m^\phi \Big) \cdot (A - \partial \phi) - j_m \cdot B + \mathcal{L}_G \, , \qquad (5.12)$$

where we denoted the part of the magnetic current $j_m$ carrying an instanton degree of freedom $\phi$ by $j_m^\phi$. For instance, $j_m^\phi$ can correspond to a current of 't Hooft-Polyakov monopoles. Let us remind the reader that $\mathcal{L}_G$ is the gauge-fixing Lagrangian given by Eq. (4.5), $\bar{j}_e$ is the part of the electric current which is quantized in units of elementary electric charge and $y_+$, $\alpha_1$, $\beta_1$, $a_-$, $b_-$, $y_-$ are the QEMD operators defined in sec. 4.3. Note that since we derived the Lagrangian (5.12) from general symmetry arguments, the field $a$ entering our EFT need not be the QCD axion, but could as well correspond to a generic ALP.

Let us derive the classical equations of motion corresponding to the Lagrangian (5.12). For this, we follow the procedure outlined in secs. 3.4.1 and 4.2. Varying over the two

four-potentials, we obtain:

$$\int\limits_{\Sigma} d\Sigma_\mu \left\{ \frac{n\cdot\partial}{n^2} \left( n\cdot\partial A^\mu - \partial^\mu n\cdot A - n^\mu \partial\cdot A - \epsilon^\mu{}_{\nu\rho\sigma} n^\nu \partial^\rho B^\sigma \right) \right.$$

$$\left. - g_{a\mathrm{AA}}\, \partial_\nu a \left\{ (\partial\wedge A)^d \right\}^{\nu\mu} - g_{a\mathrm{AB}}\, \partial_\nu a \left\{ (\partial\wedge B)^d \right\}^{\nu\mu} - \bar{j}_e^\mu - g_{a\mathrm{Aj}}\, a\, j_m^{\phi\,\mu} \right\} = 0 \,, \tag{5.13}$$

$$\int\limits_{\Sigma} d\Sigma_\mu \left\{ \frac{n\cdot\partial}{n^2} \left( n\cdot\partial B^\mu - \partial^\mu n\cdot B - n^\mu \partial\cdot B - \epsilon^\mu{}_{\nu\rho\sigma} n^\nu \partial^\rho A^\sigma \right) \right.$$

$$\left. - g_{a\mathrm{BB}}\, \partial_\nu a \left\{ (\partial\wedge B)^d \right\}^{\nu\mu} - g_{a\mathrm{AB}}\, \partial_\nu a \left\{ (\partial\wedge A)^d \right\}^{\nu\mu} - j_m^\mu \right\} = 0 \,, \tag{5.14}$$

where $\Sigma$ is an arbitrary 3-surface of the space-time manifold $M$. Note that the integral form of the equations is essential as discussed in sec. 3.4.1. Then, transitioning to the description in terms of the field strength tensor $F$, we find the following axion Maxwell equations:

$$\int\limits_{\Sigma} d\Sigma_\nu \left( \partial_\mu F^{\mu\nu} - g_{a\mathrm{AA}}\, \partial_\mu a\, F^{d\,\mu\nu} + g_{a\mathrm{AB}}\, \partial_\mu a\, F^{\mu\nu} - \bar{j}_e^\nu - g_{a\mathrm{Aj}}\, a\, j_m^{\phi\,\nu} \right) = 0\,, \tag{5.15}$$

$$\int\limits_{\Sigma} d\Sigma_\nu \left( \partial_\mu F^{d\,\mu\nu} + g_{a\mathrm{BB}}\, \partial_\mu a\, F^{\mu\nu} - g_{a\mathrm{AB}}\, \partial_\mu a\, F^{d\,\mu\nu} - j_m^\nu \right) = 0\,. \tag{5.16}$$

Note that the terms proportional to $(n\cdot\partial)^{-1}\,(n\wedge j_m)^{\mu\nu}$ and $(n\cdot\partial)^{-1}\,(n\wedge j_e)^{\mu\nu}$ do not contribute to the classical equations of motion, as it was discussed in sec. 4.2, see also a rigorous derivation within the path integral approach in Ref. [81][15]. The IR electromagnetic fields $F_{\mu\nu}$ satisfy homogeneous differential axion Maxwell equations:

$$\begin{cases} \partial_\mu F^{\mu\nu} - g_{a\mathrm{AA}}\, \partial_\mu a\, F^{d\,\mu\nu} + g_{a\mathrm{AB}}\, \partial_\mu a\, F^{\mu\nu} = 0 \\ \partial_\mu F^{d\,\mu\nu} + g_{a\mathrm{BB}}\, \partial_\mu a\, F^{\mu\nu} - g_{a\mathrm{AB}}\, \partial_\mu a\, F^{d\,\mu\nu} = 0 \end{cases}\,, \quad x \in M\backslash\{\mathcal{S}_i\}\,, \tag{5.17}$$

since their support has to be restricted to all points of the space-time manifold $M$ apart from the worldlines of charged particles $\{\mathcal{S}_i\}$, see secs. 3.3 and 3.4.1. Equations (5.17) guarantee that the low energy continuity laws for the electric and magnetic currents are violated only for the current of the 't Hooft-Polyakov monopoles $j_m^{\phi\,\nu}$ (or any other magnetically charged objects carrying an extra rotor degree of freedom $\phi$), as it was discussed in sec. 3.4.2. As there are no such monopoles found, we set $j_m^\phi = 0$ in the further discussion for simplicity.

---

[15]That such terms cannot contribute to the classical equations of motion is also clear from the fact that the interaction of axions with the electromagnetic field cannot depend on the kind of currents sourcing the latter field: for instance, in a given setting the axion field could be causally disconnected from these currents. In fact, one can obtain the axion Maxwell equations (5.19), (5.20) and (5.18) by using an even simpler two-potential framework of Ref. [94] which does not involve currents at all.

Eqs. (5.15) and (5.16) are to be supplemented by the following equation of motion for the axion field:

$$\left(\partial^2 - m_a^2\right) a \;=\; \frac{1}{4}\left(g_{a\mathrm{AA}} - g_{a\mathrm{BB}}\right) F_{\mu\nu} F^{d\,\mu\nu} - \frac{1}{2}\, g_{a\mathrm{AB}} F_{\mu\nu} F^{\mu\nu}, \qquad (5.18)$$

where the right-hand side is obtained by varying the Lagrangian (5.12) with respect to the axion field and transitioning to the description in terms of the field strength tensor $F$. According to the discussion of the previous section, the axion mass $m_a$ receives an additional contribution from the Witten-effect induced interaction in the case where there exist monopoles with the instanton degrees of freedom $\phi$.

Let us now bring Eqs. (5.15), (5.16) and (5.18) into the form convenient for their experimental study. First, we set $j_m = 0$, since there are no magnetic monopoles in the laboratory. Second, we expand the electromagnetic field in powers of the anomalous axion-photon couplings $g_{a\mathrm{AA}}$, $g_{a\mathrm{BB}}$ and $g_{a\mathrm{AB}}$, keeping only the zeroth and the first orders $F = F_0 + F_a$. At zeroth order, Eqs. (5.15), (5.16) and (5.18) decouple and give the ordinary Maxwell equations as well as the homogeneous Klein-Gordon equation for the axion field. At first order, Eqs. (5.15) and (5.16) yield:

$$\partial_\mu F_a^{\mu\nu} - g_{a\mathrm{AA}}\, \partial_\mu a\, F_0^{d\,\mu\nu} + g_{a\mathrm{AB}}\, \partial_\mu a\, F_0^{\mu\nu} \;=\; 0\,, \qquad (5.19)$$

$$\partial_\mu F_a^{d\,\mu\nu} + g_{a\mathrm{BB}}\, \partial_\mu a\, F_0^{\mu\nu} - g_{a\mathrm{AB}}\, \partial_\mu a\, F_0^{d\,\mu\nu} \;=\; 0\,, \qquad (5.20)$$

so that $F_a$ is an axion-induced part of the electromagnetic field sourced by the following effective electric and magnetic currents:

$$j_{e,\,\mathrm{eff}}^\nu = g_{a\mathrm{AA}}\, \partial_\mu a\, F_0^{d\,\mu\nu} - g_{a\mathrm{AB}}\, \partial_\mu a\, F_0^{\mu\nu}\,, \qquad (5.21)$$

$$j_{m,\,\mathrm{eff}}^\nu = -g_{a\mathrm{BB}}\, \partial_\mu a\, F_0^{\mu\nu} + g_{a\mathrm{AB}}\, \partial_\mu a\, F_0^{d\,\mu\nu}\,, \qquad (5.22)$$

which depend on the external field $F_0$ created in the laboratory. Eqs. (5.21) and (5.22) extend the results of the axion EFT of Ref. [95], which yields $j_{e,\,\mathrm{eff}}^\nu = g_{a\mathrm{AA}}\, \partial_\mu a\, F_0^{d\,\mu\nu}$ and $j_{m,\,\mathrm{eff}}^\nu = 0$. As we discussed in sec. 5.1, in an axion model with a generic spectrum of heavy PQ-charged dyons the couplings satisfy $g_{a\mathrm{BB}} \gg |g_{a\mathrm{AB}}| \gg g_{a\mathrm{AA}}$, so that the additional terms we obtain dominate the conventional contribution to the effective currents.

Finally, in terms of electric and magnetic fields, Eqs. (5.18), (5.19) and (5.20) are given by the following expressions:

$$\boldsymbol{\nabla}\times\mathbf{B}_a - \dot{\mathbf{E}}_a = g_{a\mathrm{AA}}\left(\mathbf{E}_0\times\boldsymbol{\nabla} a - \dot{a}\mathbf{B}_0\right) + g_{a\mathrm{AB}}\left(\mathbf{B}_0\times\boldsymbol{\nabla} a + \dot{a}\mathbf{E}_0\right), \qquad (5.23)$$

$$\boldsymbol{\nabla}\times\mathbf{E}_a + \dot{\mathbf{B}}_a = -g_{a\mathrm{BB}}\left(\mathbf{B}_0\times\boldsymbol{\nabla} a + \dot{a}\mathbf{E}_0\right) - g_{a\mathrm{AB}}\left(\mathbf{E}_0\times\boldsymbol{\nabla} a - \dot{a}\mathbf{B}_0\right), \qquad (5.24)$$

$$\boldsymbol{\nabla}\cdot\mathbf{B}_a = -g_{a\mathrm{BB}}\,\mathbf{E}_0\cdot\boldsymbol{\nabla} a + g_{a\mathrm{AB}}\,\mathbf{B}_0\cdot\boldsymbol{\nabla} a\,, \qquad (5.25)$$

$$\boldsymbol{\nabla}\cdot\mathbf{E}_a = g_{a\mathrm{AA}}\,\mathbf{B}_0\cdot\boldsymbol{\nabla} a - g_{a\mathrm{AB}}\,\mathbf{E}_0\cdot\boldsymbol{\nabla} a\,, \qquad (5.26)$$

$$\left(\partial^2 - m_a^2\right) a = -\left(g_{a\mathrm{AA}} - g_{a\mathrm{BB}}\right)\mathbf{E}\cdot\mathbf{B} + g_{a\mathrm{AB}}\left(\mathbf{E}^2 - \mathbf{B}^2\right), \qquad (5.27)$$

where $\mathbf{E}_a$ and $\mathbf{B}_a$ are axion-induced electric and magnetic fields, while $\mathbf{E}_0$ and $\mathbf{B}_0$ are background electric and magnetic fields created in the detector. Note that to study the

propagation of light with Eqs. (5.23)–(5.27), it is convenient not to perform the expansion of the electromagnetic fields $\mathbf{E}_\gamma$ and $\mathbf{B}_\gamma$, in which case the equations are the same, but with the substitutions $\mathbf{E}_a$, $\mathbf{E}_0 \to \mathbf{E}_\gamma$ and $\mathbf{B}_a$, $\mathbf{B}_0 \to \mathbf{B}_\gamma$.

# 6 Targets for axion search experiments

## 6.1 General lessons

Let us discuss the phenomenology of the new electromagnetic couplings of axions and ALPs found in the previous sections. In our discussion, we will always consider first the general case of ALPs, i.e. Nambu-Goldstone bosons of an arbitrary spontaneously broken global $U(1)$ symmetry, which by definition include QCD axions as a special case, and only then make quantitative predictions in particular axion models.

Due to the scaling of the new $g_{a\mathrm{BB}}$ and $g_{a\mathrm{AB}}$ couplings with the elementary electric and magnetic charge units found in sec. 5.1, in any model where $g_{a\mathrm{AB}} \neq 0$, one expects the ratio $g_{a\mathrm{BB}}/|g_{a\mathrm{AB}}|$ to be proportional to a large number $g_0/e \gg 1$. This means that the possible effects associated to the $g_{a\mathrm{AB}}$ coupling play the dominant role only for those observables, which do not get any sizable contribution from the $g_{a\mathrm{BB}}$ coupling. As we will discuss in the next sections, such observables do exist and can be probed in various experiments by studying the interactions of ALPs with polarized light, searching for EDMs of charged particles and a fifth force or by looking for light ALP dark matter in an external magnetic field with haloscopes.

Still, for most of the processes involving ALP-photon interactions, the dominant effect is associated to the $g_{a\mathrm{BB}}$ coupling. Symmetry of Eq. (5.27) with respect to the interchange of the $g_{a\mathrm{AA}}$ and $g_{a\mathrm{BB}}$ couplings (up to an insignificant sign) suggests that in any process of axion production by the electromagnetic fields, the effect of the $g_{a\mathrm{BB}}$ coupling is analogous to the effect of the conventional $g_{a\gamma\gamma}$ coupling [16]. Moreover, in the case of relativistic axions propagating perpendicular to the external magnetic (electric) field, Eqs. (5.23)–(5.24) show that the effective electric and magnetic axion-induced currents differ only by the coefficients $g_{a\mathrm{AA}}$ or $g_{a\mathrm{BB}}$, respectively, which means that the power in axion-induced electromagnetic fields is similar up to a coefficient. Thus, the rates of such processes as conversion of the relativistic ALPs into photons and back in transverse magnetic fields [96], and ALP emission through Primakoff effect [97] or photon coalescence, are all given by the conventional expressions, but with the $g_{a\gamma\gamma}$ coupling substituted by the $g_{a\mathrm{BB}}$ one.

The same simple rule of replacing the $g_{a\gamma\gamma}$ coupling with the $g_{a\mathrm{BB}}$ coupling in conventional expressions applies to the dispersion relation of light in an ALP background and to the ALP decay to two photons. Indeed, after we omit the subdominant $|g_{a\mathrm{AB}}| \ll g_{a\mathrm{BB}}$ coupling and put $\mathbf{E}_a$, $\mathbf{E}_0 \to \mathbf{E}_\gamma$ and $\mathbf{B}_a$, $\mathbf{B}_0 \to \mathbf{B}_\gamma$, the axion Maxwell equations (5.23)–(5.26) become invariant under the interchange of the couplings $g_{a\mathrm{AA}}$ and $g_{a\mathrm{BB}}$ supplemented by the electric-magnetic duality transformation $\mathbf{E}_\gamma \to \mathbf{B}_\gamma$, $\mathbf{B}_\gamma \to -\mathbf{E}_\gamma$. Since the propagation of light is electric-magnetic duality invariant, the $g_{a\mathrm{BB}}$ coupling enters the dispersion relation and the decay width in the same way as the conventional $g_{a\mathrm{AA}}$ coupling. It can also be

---

[16]This statement need not hold for loop effects, as Eq. (5.27) is classical.

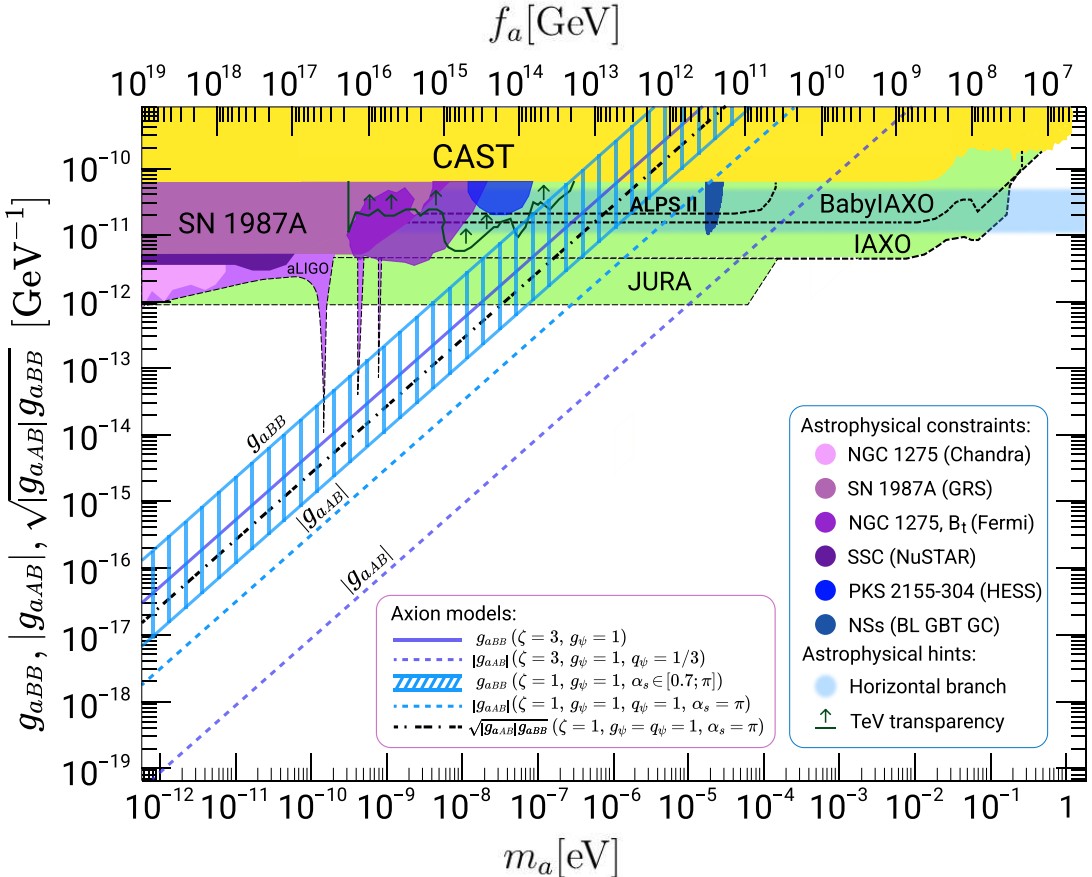

**Figure 1.** Existing and projected (dashed lines) constraints on the parameter space of ALP-photon $g_{a\mathrm{BB}}$ and $g_{a\mathrm{AB}}$ couplings versus ALP mass and decay constant together with the lines corresponding to $g_{a\mathrm{BB}}$ (solid), $|g_{a\mathrm{AB}}|$ (dashed) and $\sqrt{|g_{a\mathrm{AB}}|\,g_{a\mathrm{BB}}}$ (dash-dotted) in different hadronic axion models with one heavy PQ-charged fermion $\psi$ with the parameters given in a box and $N_{\mathrm{DW}} = 1$. Astrophysical hints are also shown. For further discussion, see main text.

explicitly checked that the form of the second-order differential equations for $\mathbf{E}_\gamma$ and $\mathbf{B}_\gamma$ does not change.

Let us consider the existing constraints on the ALP-photon $g_{a\gamma\gamma}$ coupling which take advantage of the effects discussed in the previous two paragraphs. It is now clear that the same constraints hold also for the new $g_{a\mathrm{BB}}$ coupling and the corresponding search strategies need not be updated. In particular, this is the case of helioscope searches [98–100], light-shining-through-wall (LSW) [101–107] and axion interferometry [108–112] experiments, as well as many astrophysical and cosmological constraints [65]. We present the corresponding constraints on the $g_{a\mathrm{BB}}$ coupling in Fig. 1. Note that as $|g_{a\mathrm{AB}}| \ll g_{a\mathrm{BB}}$, the same constraints obviously hold for $|g_{a\mathrm{AB}}|$ and $\sqrt{|g_{a\mathrm{AB}}|\,g_{a\mathrm{BB}}}$.

The qualitative distinction between the new $g_{a\mathrm{BB}}$ coupling and the conventional $g_{a\gamma\gamma}$ coupling arises whenever a given process cannot be described by Eq. (5.27) and involves observables which are not invariant under the electric-magnetic duality symmetry. In this

case, the values of these observables derived from the axion Maxwell equations (5.23)–(5.26) are not symmetric with respect to the interchange $g_{aAA} \leftrightarrow g_{aBB}$ of the two couplings, so that there is a qualitative difference in the effects of these couplings. We give a particular example where the latter difference plays a crucial role in sec. 6.3. In particular, in the latter section, we discuss haloscope experiments searching for light ALP dark matter ($m_a \ll \mu$eV). We find that in this case, the constraints obtained for the $g_{a\gamma\gamma}$ coupling need not hold for the $g_{aBB}$ coupling. This means that to probe the latter coupling, these experiments should exploit a search strategy which is different from the one normally used.

To be more specific, when we make quantitative predictions in the next sections, we will consider a particular kind of ALP: the QCD axion. In this case, we will take advantage of Eqs. (5.3), (5.4) and (5.6) for the axion-photon couplings. As we discussed in sec. 5.1, there are two families of axion models where the new electromagnetic couplings $g_{aAB}$ and $g_{aBB}$ can arise: those with Abelian ($\zeta = 3$) and those with non-Abelian ($\zeta = 1$) magnetic monopoles, cf. Eq. (5.7). In the former case, the axion decay constant $f_a$ is obviously related to the QCD anomaly coefficient $N$ and PQ scale $v_a$ in the standard way: $f_a = v_a/2N$, while in the latter case, the relation is non-standard [18]: $f_a = 2\alpha_s^2 v_a/N$, where $\alpha_s = g_s^2/4\pi$. Using these relations, we plot the lines corresponding to $g_{aBB}$, $|g_{aAB}|$ and $\sqrt{|g_{aAB}| g_{aBB}}$ [17] in different hadronic axion models. For simplicity, we choose the models having only one heavy vector-like PQ-charged fermion $\psi$, which transforms trivially under the $SU(2)_L$ gauge group of the weak interactions and in the fundamental representation under the color $SU(3)_c$ gauge group (electric $SU(3)_E$ in the Abelian monopole case or magnetic $SU(3)_M$ in the non-Abelian monopole case), and has charges $q_\psi$ and $g_\psi$, see Fig. 1 and the legend therein. In these models, the QCD anomaly coefficient is $N = 1/2$, so that $N_{DW} = 1$. In the non-Abelian monopole case $\zeta = 1$, there is an uncertainty associated to our ignorance of the exact value of $\alpha_s$ at low energies [113]. Note that the axion models populate a region of the parameter space, which in the analogous plot for the $g_{aAA}$ coupling would be extensively probed by existing and projected haloscope experiments searching for light ALP dark matter ($m_a < \mu$eV). However, as we discussed briefly in the previous paragraph and will elaborate later in sec. 6.3, the constraints from such haloscopes cannot be translated to Fig. 1. Also, note that the conventional KSVZ and DFSZ axion lines are obviously missing from the plot in Fig. 1, since this plot depicts the $g_{aBB}$ and $|g_{aAB}|$ couplings, but not the $g_{aAA}$ coupling.

## 6.2 Purely laboratory-based experiments

A particularly clean way to measure the $g_{aAB}$ coupling is provided by LSW experiments [114]. As one can see from Eq. (5.27), contrary to the CP-conserving couplings, the CP-violating $g_{aAB}$ coupling allows interaction between ALPs and light polarized in a plane perpendicular to the external magnetic field. As one can control the polarization of the incoming light in a LSW experiment, it is straightforward to artificially turn off the CP-conserving ALP-photon interaction on the photon to ALP conversion side before the wall. Using $g_{aBB} \gg |g_{aAB}| \gg g_{aAA}$, we find the following LSW probabilities corresponding to different

---

[17]The $\sqrt{|g_{aAB}| g_{aBB}}$ line is relevant for LSW searches, see Eq. (6.2) and discussion in sec. 6.2.

linear polarizations of incoming light:

$$P(\gamma_\parallel \to a \to \gamma) \simeq 16 \, \frac{(g_{a\mathrm{BB}}\omega B_0)^4}{m_a^8} \, \sin^4\!\left(\frac{m_a^2 L_{B_0}}{4\omega}\right), \tag{6.1}$$

$$P(\gamma_\perp \to a \to \gamma) \simeq 16 \, \frac{(g_{a\mathrm{AB}}\omega B_0)^2 (g_{a\mathrm{BB}}\omega B_0)^2}{m_a^8} \, \sin^4\!\left(\frac{m_a^2 L_{B_0}}{4\omega}\right), \tag{6.2}$$

where $\gamma_\parallel$ ($\gamma_\perp$) denotes the incoming light with frequency $f = \omega/(2\pi)$ and with polarisation parallel (perpendicular) to the magnetic field $B_0$, which is supposed to be transverse to the direction of the light beam and which is sustained in a cavity of length $L_{B_0}$, both before and behind the wall. Clearly, from a detection of LSW with some probability $P(\gamma_\parallel \to a \to \gamma)$, one can determine the $g_{a\mathrm{BB}}$ coupling in a first measurement. In a second measurement, one can also search for LSW via $\gamma_\perp \to a \to \gamma$. Then, for the case of axions, using Eqs. (5.4), (5.6) and (5.7), we see that the coupling $g_{a\mathrm{AB}}$ can be determined from the following ratio:

$$\frac{P(\gamma_\perp \to a \to \gamma)}{P(\gamma_\parallel \to a \to \gamma)} \simeq \frac{g_{a\mathrm{AB}}^2}{g_{a\mathrm{BB}}^2} = \left(\frac{D}{M}\frac{e}{g_0}\right)^2 = 4\left(\frac{D}{\zeta M}\right)^2 \alpha^2 \simeq 2.13 \times 10^{-4} \left(\frac{D}{\zeta M}\right)^2. \tag{6.3}$$

For example, the experiment ALPS II ($B_0 = 5.3\,\mathrm{T}$, $L_{B_0} = 105.6\,\mathrm{m}$, $\omega = 1.17\,\mathrm{eV}$) has the capability to search for LSW using incoming light with both polarisations, $\gamma_\parallel$ and $\gamma_\perp$ [115]. For both of them, ALPS II has the projected sensitivity $P_{\mathrm{sens}} \approx 10^{-33}$ to the corresponding LSW probabilities. This would allow for the detection of a light, $m_a \lesssim 10^{-4}\,\mathrm{eV}$, axion featuring a CP-conserving coupling $|g_{a\mathrm{AA}} - g_{a\mathrm{BB}}| \simeq |g_{a\mathrm{BB}}| \gtrsim 2 \times 10^{-11}\,\mathrm{GeV}^{-1}$ via $\gamma_\parallel \to a \to \gamma$, as can be inferred from Eq. (6.1). If this newly discovered axion features also a CP-violating coupling, then the latter has to be in the range $|g_{a\mathrm{AB}}| = 2\alpha(|D|/\zeta M)g_{a\mathrm{BB}} \gtrsim 3 \times 10^{-13}\,\mathrm{GeV}^{-1}(|D|/\zeta M)$. If $|D| \simeq M$, to detect such a coupling via $\gamma_\perp \to a \to \gamma$ requires a sensitivity improvement by four order of magnitudes, to $P_{\mathrm{sens}} \sim 10^{-37}$, as can be seen from Eqs. (6.1) (6.2), and (6.3). Intriguingly, such a sensitivity has been argued to be achievable by the next generation LSW experiment JURA (also known as ALPS III [116, 117]). Indeed, see Fig. 1, where we showed the $g_{a\mathrm{BB}}$ ($\sqrt{|g_{a\mathrm{AB}}| g_{a\mathrm{BB}}}$) parameter space probed by JURA according to Eq. (6.1) (Eq. (6.2)) together with the lines corresponding to $g_{a\mathrm{BB}}$ and $\sqrt{|g_{a\mathrm{AB}}| g_{a\mathrm{BB}}}$ in the axion model with $|D| = M = \zeta = 1$. Thus, our considerations show that an eventual detection of an ALP by ALPS II would strongly motivate the construction of JURA. After all, an experimental verification of Eq. (6.3) would allow a deep view into the UV, provide strong evidence for the existence of heavy dyons and even an insight into their spectrum via the ratio $|D|/\zeta M$.

Although LSW experiments can probe the $g_{a\mathrm{AB}}$ coupling, we saw that the effects of the $g_{a\mathrm{BB}}$ coupling are dominant and are expected to be discovered first. To the contrary, there exist purely-laboratory experiments which are primarily sensitive to the $g_{a\mathrm{AB}}$ coupling. These are the experiments which probe CP-violating observables, since only the $g_{a\mathrm{AB}}$ coupling violates CP. One can probe the corresponding CP-violating effects by searching for electric dipole moments of charged particles, such as electrons, protons and muons [118]. Moreover, the CP-violating axion-photon coupling can be probed in various experiments searching for fifth force or monopole-dipole axion-induced interactions [119], since one expects the $g_{a\mathrm{AB}}$ coupling to radiatively induce CP-violating interactions between axions and

charged fermions $f$ of the form $g_f a \bar{f} f$. Naively, one could argue that there exist strong constraints on the $g_{a\mathrm{AB}}$ coupling already, analogously to the constraints obtained in Ref. [120]. Note however, that although all the couplings of the ALP EFT (5.11) are small, the theory is still essentially non-perturbative, so predicting the exact values for the discussed CP-violating observables and thus inferring robust experimental constraints is not straightforward. We leave this task for future work.

## 6.3 Haloscope experiments

In this section, let us focus on the case where ALPs constitute dark matter and have large Compton wavelengths $\lambda_a$ compared to the length scale $L$ of the experiment[18]. In particular, this is the case of light cosmic ALPs with masses $m_a \ll \mu\mathrm{eV}$, which one aims to detect with such haloscope experiments as ABRACADABRA [123, 124], ADMX SLIC [125], DM Radio [126], SHAFT [127], WISPLC [128], and others. In these experiments, one maintains a constant magnetic field $\mathbf{B}_0$ in a laboratory and searches for an ALP-induced oscillating magnetic field $\mathbf{B}_a$. Note that due to the condition $\lambda_a \gg L$, interactions of ALPs with the field $\mathbf{B}_0$ cannot be described as a conventional ALP-photon conversion phenomenon. To determine the expected magnitude of the induced fields in this case, one has to use the axion Maxwell equations (5.23)–(5.26). The latter equations can be significantly simplified since most of the terms on the right-hand side are normally suppressed. Indeed, considering the case of axions and assuming $E \simeq M \simeq |D|$, the axion-photon couplings satisfy $g_{a\mathrm{AA}}/g_{a\mathrm{BB}} \simeq (e/g_0)^2 \lesssim 2 \cdot 10^{-4}$ and $|g_{a\mathrm{AB}}|/g_{a\mathrm{BB}} \simeq e/g_0 \lesssim 10^{-2}$. Moreover, the cosmic axions that form dark matter have typical velocities $v_a \sim 10^{-3}$, so that the gradient of the oscillating axion field is suppressed with respect to its time derivative: $|\boldsymbol{\nabla} a| \sim 10^{-3} \dot{a}$.

Leaving only the first three dominant terms, we obtain the following simplified axion Maxwell equations:

$$\boldsymbol{\nabla} \times \mathbf{B}_a - \dot{\mathbf{E}}_a = 0, \tag{6.4}$$

$$\boldsymbol{\nabla} \times \mathbf{E}_a + \dot{\mathbf{B}}_a = -g_{a\mathrm{BB}} \left( \mathbf{B}_0 \times \boldsymbol{\nabla} a + \dot{a} \mathbf{E}_0 \right) + g_{a\mathrm{AB}} \dot{a} \mathbf{B}_0, \tag{6.5}$$

$$\boldsymbol{\nabla} \cdot \mathbf{B}_a = 0, \tag{6.6}$$

$$\boldsymbol{\nabla} \cdot \mathbf{E}_a = 0, \tag{6.7}$$

where we included the dominant effect arising from an external electric field $\mathbf{E}_0$. Note that all the existing haloscopes use an external magnetic field $\mathbf{B}_0$ instead, partly because the dominant effect for the usually considered $g_{a\mathrm{AA}}$ coupling is due to the term with an external magnetic, but not electric, field and partly because it is technologically challenging to sustain a large enough electric field in a big enough volume. It is clear from Eq. (6.5) that if the latter technological problem is solved, so that $E_0 \gtrsim 10^{-2} \left( |D|/\zeta M \right) B_0$ in the CP-violating case or $E_0 \gtrsim 10^{-3} B_0$ in the CP-conserving case, the $g_{a\mathrm{BB}} \dot{a} \mathbf{E}_0$ term will allow one to search for dark matter axions in an external electric field with the sensitivity which

---

[18]Note that the case $\lambda_a \lesssim L$ is of less relevance for the simplest QCD axion models due to the existing helioscope constraints, see Fig. 1. Still, this case is very important for generic ALP dark matter, and so it was investigated in Refs. [121] and [122] while this article was being prepared for publication.

is not worse than the one of the conventional searches conducted in an external magnetic field.

Returning to the case of the existing haloscopes where $\mathbf{E}_0 = 0$, $\mathbf{B}_0 \neq 0$, we see that the axion Maxwell equations (6.4)–(6.7) have significantly different structure compared to the conventional axion Maxwell equations which take into account solely the $g_{aAA}$ coupling. While in the latter case axions generate an effective electric current $\mathbf{j}_{\mathrm{eff}}^e = -g_{aAA}\dot{a}\mathbf{B}_0$, in the former case an effective magnetic current is generated:

$$\mathbf{j}_{\mathrm{eff}}^m = g_{aBB}\mathbf{B}_0 \times \boldsymbol{\nabla} a - g_{aAB}\dot{a}\mathbf{B}_0 \,. \tag{6.8}$$

Note that in the case $M \simeq |D|$, the term proportional to the CP-violating $g_{aAB}$ coupling is dominant. To the contrary, if the underlying UV theory is CP-conserving, then $D = 0$ and $g_{aAB} = 0$, so that only the term proportional to the $g_{aBB}$ coupling contributes.

Let us now find experimental implications of the magnetic current (6.8). Applying the curl differential operator to the Eqs. (6.4) and (6.5), and using the other equations from the system (6.4)–(6.7), we obtain:

$$\Delta \mathbf{E}_a - \ddot{\mathbf{E}}_a = \boldsymbol{\nabla} \times \mathbf{j}_{\mathrm{eff}}^m \,, \tag{6.9}$$

$$\Delta \mathbf{B}_a - \ddot{\mathbf{B}}_a = \partial \mathbf{j}_{\mathrm{eff}}^m / \partial t \,. \tag{6.10}$$

The leading terms contributing to the right-hand side are:

$$\boldsymbol{\nabla} \times \mathbf{j}_{\mathrm{eff}}^m = g_{aBB}\left(\boldsymbol{\nabla} a \cdot \boldsymbol{\nabla}\right)\mathbf{B}_0 - g_{aAB}\dot{a}\,\boldsymbol{\nabla} \times \mathbf{B}_0 \,, \tag{6.11}$$

$$\partial \mathbf{j}_{\mathrm{eff}}^m / \partial t = g_{aBB}\mathbf{B}_0 \times \boldsymbol{\nabla}\dot{a} - g_{aAB}\ddot{a}\mathbf{B}_0 \,. \tag{6.12}$$

The axion dark matter field is given by the following expression: $a(t, \mathbf{r}) = a_0 \cos(\omega_a t - \mathbf{k}_a \cdot \mathbf{r})$, where $\omega_a = m_a$ and $\mathbf{k}_a = m_a \mathbf{v}_a$. The leading CP-conserving effect then depends on the direction of the axion wind and thus experiences modulations with the periods of one sidereal day $T_d$ and one sidereal year $T_y$ due to the rotation of the Earth around its axis and around the Sun, respectively. To find the axion-induced $\mathbf{E}_a$ and $\mathbf{B}_a$ fields, Eqs. (6.9) and (6.10) have to be solved for a particular geometry of a given haloscope experiment.

Let us illustrate the general features of the solution by considering the example of a very long solenoid of radius $R$ with magnetic field $\mathbf{B}_0$ directed along the z-axis. In this case, Eqs. (6.9) and (6.10) become:

$$\Delta \mathbf{E}_a - \ddot{\mathbf{E}}_a = -\left(g_{aBB}\,\partial_\rho a\,\mathbf{e}_z + g_{aAB}\dot{a}\,\mathbf{e}_\phi\right)B_0\,\delta(\rho - R) \,, \tag{6.13}$$

$$\Delta \mathbf{B}_a - \ddot{\mathbf{B}}_a = g_{aBB}\mathbf{B}_0 \times \boldsymbol{\nabla}\dot{a} - g_{aAB}\ddot{a}\mathbf{B}_0 \,, \tag{6.14}$$

where we work in cylindrical coordinates $(\rho, \phi, z)$ with unit vectors $(\mathbf{e}_\rho, \mathbf{e}_\phi, \mathbf{e}_z)$. Let us parameterize the direction of the axion wind $\hat{\boldsymbol{v}}_a = (\sin\theta\cos(\phi - \xi), -\sin\theta\sin(\phi - \xi), \cos\theta)$ in cylindrical coordinates by two angles $\theta$ and $\xi$. Assuming $T_d \gg 2\pi/\omega_a$, which corresponds to $m_a \gg 5 \cdot 10^{-20}$ eV, we neglect the terms proportional to $\dot{\xi}$ and $\dot{\theta}$ in the Eqs. (6.13) and (6.14). It is then straightforward to obtain the solutions of these equations in terms of Bessel functions. All we need however are these solutions in the limit $\omega_a R \ll 1$, as we

are interested in axions with large Compton wavelengths. In the latter limit, solutions to Eqs. (6.13), (6.14) with physical boundary conditions are:

$$
\mathbf{E}_a = \begin{cases}
\frac{1}{2}\, a_0\, \omega_a \rho\, B_0 \Big( g_{a\mathrm{AB}}\, \mathbf{e}_\phi - g_{a\mathrm{BB}}\, v_a \sin\theta\, \cos(\phi - \xi)\, \mathbf{e}_z \Big) \sin\omega_a t\,, \quad \rho < R \\[2mm]
\frac{1}{2}\, a_0\, \omega_a \frac{R^2}{\rho}\, B_0 \Big( g_{a\mathrm{AB}}\, \mathbf{e}_\phi - g_{a\mathrm{BB}}\, v_a \sin\theta\, \cos(\phi - \xi)\, \mathbf{e}_z \Big) \sin\omega_a t\,, \quad \rho > R
\end{cases} , \qquad (6.15)
$$

$$
\mathbf{B}_a = \begin{cases}
\frac{1}{2}\, a_0\, (\omega_a R)^2\, B_0 \Big( g_{a\mathrm{AB}}\, \mathbf{e}_z + g_{a\mathrm{BB}}\, v_a \sin\theta \big\{ \cos(\phi - \xi)\, \mathbf{e}_\phi + \\
\qquad\qquad\qquad \sin(\phi - \xi)\, \mathbf{e}_\rho \big\} \Big) \Big( \ln\omega_a R + \frac{\rho^2}{2R^2} \Big) \cos\omega_a t\,, \quad \rho < R \\[2mm]
\frac{1}{2}\, a_0\, (\omega_a R)^2\, B_0 \Big( g_{a\mathrm{AB}}\, \mathbf{e}_z + g_{a\mathrm{BB}}\, v_a \sin\theta \big\{ \cos(\phi - \xi)\, \mathbf{e}_\phi + \\
\qquad\qquad\qquad \sin(\phi - \xi)\, \mathbf{e}_\rho \big\} \Big) \ln\omega_a \rho\, \cos\omega_a t\,, \quad \rho > R
\end{cases} . \qquad (6.16)
$$

It is immediately clear that $E_a \gg B_a$, which means that in an experiment with the geometry of our example one has to search for axion-induced electric, but not magnetic, fields. Moreover, it turns out that this feature persists for any other possible geometry as well. Indeed, our Eqs. (6.9) and (6.10) can be rendered equivalent to the equations studied in Ref. [129] by substituting $\mathbf{E}_a \to \mathbf{B}_a$, $\mathbf{B}_a \to -\mathbf{E}_a$ and $\mathbf{j}_{\mathrm{eff}}^m \to \mathbf{j}_{\mathrm{eff}}^e$. In the latter work, it was found that for any haloscope geometry with characteristic length scale $L$, the equation involving the time derivative of the effective current yields solutions which are suppressed by powers of $\omega_a L \ll 1$ with respect to the solutions of the equation involving the curl of this current. In our case, this means that in any haloscope probing $m_a \ll \mu\mathrm{eV}$, the axion-induced magnetic field $\mathbf{B}_a$ is suppressed with respect to the axion-induced electric field $\mathbf{E}_a$.

Note however that all the existing as well as many projected haloscopes searching for such light axions – such as ABRACADABRA, ADMX SLIC, DM Radio, SHAFT, ... – aim to measure only the axion-induced magnetic, but not electric, fields. Thus, the constraints on the conventional $g_{a\mathrm{AA}}$ coupling obtained by these experiments do not hold for the dominant axion-photon couplings $g_{a\mathrm{AB}}$ and $g_{a\mathrm{BB}}$. The latter couplings can be probed by future haloscopes equipped with electric field sensors[19]. One haloscope of such kind has already been proposed, see Refs. [131, 132]. We hope that our work will encourage more experimental effort in this direction, as we provided a sound theoretical motivation for such an endeavor. Since the electromagnetic fields generated in a haloscope by the $g_{a\mathrm{BB}}$ and $g_{a\mathrm{AB}}$ couplings are qualitatively different from the fields generated by the conventional $g_{a\mathrm{AA}}$ coupling, for which $B_a \gg E_a$ [129], the first detection of cosmic axions with electric, but not magnetic, sensor haloscope would not only constitute the discovery of axions and dark matter, but also provide a circumstantial experimental evidence for the existence of heavy magnetically charged particles. Furthermore, as one can see from Eq. (6.15), the direction of the detected electric field could allow one to infer the ratio $g_{a\mathrm{AB}}/g_{a\mathrm{BB}} = 2\alpha(D/\zeta M)$ and thus get information about the spectrum of dyons in the UV.

---

[19]While this work was being prepared for publication, the new search strategies were discussed in more detail in Ref. [130].

| | Axions | Gravitational waves |
|---|---|---|
| $\mathbf{P}_e$ | $-g_{a\mathrm{AA}}\, a\mathbf{B}_0 + g_{a\mathrm{AB}}\, a\mathbf{E}_0$ | $\mathbf{h}\times\mathbf{B}_0 + H\mathbf{E}_0 - \left(h_{00} + \frac{1}{2}\, h\right)\mathbf{E}_0$ |
| $\mathbf{M}_m$ | $g_{a\mathrm{BB}}\, a\mathbf{B}_0 + g_{a\mathrm{AB}}\, a\mathbf{E}_0$ | |
| $\mathbf{M}_e$ | $-g_{a\mathrm{AA}}\, a\mathbf{E}_0 - g_{a\mathrm{AB}}\, a\mathbf{B}_0$ | $-\mathbf{h}\times\mathbf{E}_0 + H\mathbf{B}_0 - \left(h_{00} + \frac{1}{2}\, h\right)\mathbf{B}_0$ |
| $\mathbf{P}_m$ | $g_{a\mathrm{BB}}\, a\mathbf{E}_0 - g_{a\mathrm{AB}}\, a\mathbf{B}_0$ | |

**Table 1**. Comparison between axion and gravitational wave electrodynamics in external electric $\mathbf{E}_0$ and magnetic $\mathbf{B}_0$ fields in terms of the effective induced polarization and magnetization vectors, defined by Eqs. (6.17)–(6.20); $\mathbf{h}$, $H$, $h_{00}$ and $h$ are characteristics of the gravitational wave defined in Ref. [133].

Finally, to compare different extensions of electrodynamics which could be probed by haloscope experiments, it is convenient to reexpress the axion Maxwell equations in terms of the axion-induced polarization and magnetization vectors [129, 134], which are defined as follows:

$$\boldsymbol{\nabla}\times\mathbf{B}_a - \dot{\mathbf{E}}_a = \frac{\partial\mathbf{P}^e_{\mathrm{eff}}}{\partial t} + \boldsymbol{\nabla}\times\mathbf{M}^e_{\mathrm{eff}}\,, \tag{6.17}$$

$$\boldsymbol{\nabla}\times\mathbf{E}_a + \dot{\mathbf{B}}_a = -\frac{\partial\mathbf{P}^m_{\mathrm{eff}}}{\partial t} + \boldsymbol{\nabla}\times\mathbf{M}^m_{\mathrm{eff}}\,, \tag{6.18}$$

$$\boldsymbol{\nabla}\cdot\mathbf{B}_a = -\boldsymbol{\nabla}\cdot\mathbf{P}^m_{\mathrm{eff}}\,, \tag{6.19}$$

$$\boldsymbol{\nabla}\cdot\mathbf{E}_a = -\boldsymbol{\nabla}\cdot\mathbf{P}^e_{\mathrm{eff}}\,. \tag{6.20}$$

Note that along with the ordinary effective electric polarization and magnetization vectors $\mathbf{P}^e_{\mathrm{eff}}$ and $\mathbf{M}^e_{\mathrm{eff}}$, we introduced effective magnetic polarization and magnetization vectors $\mathbf{P}^m_{\mathrm{eff}}$ and $\mathbf{M}^m_{\mathrm{eff}}$, which describe electromagnetic properties of an effective medium consisting of magnetically charged particles. In Table 1, we give explicit expressions for the effective polarization and magnetization vectors corresponding to a generic axion in an external electromagnetic field. The fact that it is possible to bring the axion Maxwell equations (5.23)–(5.26) into the form (6.17)–(6.20) suggests that the effects of axions in external electromagnetic fields are analogous to the effects of a certain medium consisting of both electrically and magnetically charged particles. From what has been discussed before, it is clear that for a generic axion, effects of the magnetic polarization (magnetization) vectors $\mathbf{P}^m_{\mathrm{eff}}$ ($\mathbf{M}^m_{\mathrm{eff}}$) in an external magnetic field dominate the effects of the electric polarization (magnetization) vectors $\mathbf{P}^e_{\mathrm{eff}}$ ($\mathbf{M}^e_{\mathrm{eff}}$). Such asymmetry between the effects exhibited by electric and magnetic constituents of the effective medium is to be contrasted with the case of gravitational wave electrodynamics [133], where there exists an electric-magnetic $U(1)$ duality symmetry rendering electric and magnetic variables equivalent. This difference in the symmetry properties of the two theories can be easily understood from the fact that the axion-photon interactions are fundamentally mediated by heavy charged particles which break the duality symmetry, while in General Relativity, the interaction between gravity and electromagnetic field is direct and independent of any charges. For an experimentalist, this distinction signifies a substantial difference in the distribution of the induced

electromagnetic fields inside the haloscope. Indeed, as it was discussed before, for a generic light axion ($m_a \ll \mu$eV, $g_{a\mathrm{BB}} \gg |g_{a\mathrm{AB}}| \gg g_{a\mathrm{AA}}$) in an external magnetic field one expects $\mathbf{j}_{\mathrm{eff}}^m \gg \mathbf{j}_{\mathrm{eff}}^e$ and thus $E_a \gg B_a$, while for a gravitational wave in an external magnetic field, plugging the expressions given in Table 1 into the Eqs. (6.17)–(6.20), one obtains $\mathbf{j}_{\mathrm{eff}}^m \sim \mathbf{j}_{\mathrm{eff}}^e$ and thus $E_a \sim B_a$.

## 7 Conclusion

As it was asserted by J. Polchinski [32], "the existence of magnetic monopoles seems like one of the safest bets that one can make about physics not yet seen". Indeed, in the beginning of this article, we reviewed a number of theoretical arguments which together provide an overwhelming theoretical evidence for magnetic monopoles, like there probably exists for no other kind of hypothetical particles. There are nevertheless multiple practical challenges behind the experimental study of monopoles. While some of the previously mentioned arguments for the existence of monopoles do not restrict their masses, another part of these arguments suggests that monopoles are super heavy, with masses well beyond the energy reach of the present-day collider experiments. Thus, although monopoles almost certainly exist from the theoretical point of view, it might be difficult to directly probe them with existing and near-future experiments.

In this work, we proposed experiments which could probe magnetic monopoles indirectly, in particular through the influence virtual monopoles would exhibit on the interactions of axions with an electromagnetic field. To study the effects of virtual monopoles, we classified all independent marginal operators preserving the symmetries and degrees of freedom of QEMD. Then, we built a generic axion-photon EFT, which holds for the QCD axion as well as for other ALPs. We showed that this EFT features previously unknown couplings which arise from UV models containing magnetic charges. We then found that these new couplings give rise to unique experimental signatures in LSW experiments and in some kinds of axion searches, such as haloscopes searching for low-mass axions ($m_a \ll \mu$eV). In the latter case, we showed that the best sensitivity to the new electromagnetic couplings of axions could be achieved by measuring an induced oscillating electric field, instead of an induced oscillating magnetic field, contrary to the setup of existing experiments. The case of low mass axions is particularly interesting, since, as it can be seen from Fig. 1, the simplest axion models predict rather large $g_{a\mathrm{BB}}$ and $g_{a\mathrm{AB}}$ couplings, which are not excluded and which can be probed by many projected experiments. Thus, we encourage the development of electric sensor haloscopes which would search for axion dark matter in the corresponding parameter region.

Apart from unveiling these intriguing experimental applications, our work reconsidered several questions in axion theory. First, we showed that contrary to the existing statements, the main contribution to the axion-photon coupling need not be quantized in units proportional to $e^2$. Indeed, we established that the standard axion-photon coupling $g_{a\gamma\gamma} \equiv g_{a\mathrm{AA}}$ and the Witten-effect induced coupling $g_{a\mathrm{Aj}}$ are two different couplings and therefore the previous quantization arguments based on the Witten effect apply only to the $g_{a\mathrm{Aj}}$ interaction. While the other couplings $g_{a\mathrm{AA}}$, $g_{a\mathrm{AB}}$ and $g_{a\mathrm{BB}}$, as Chern-Simons-type

couplings, have to be quantized in the case of a non-trivial topology of the base manifold, the corresponding quantization units are proportional to $e^2$ only in the case of the $g_{a\mathrm{AA}}$ coupling, whereas the $g_{a\mathrm{AB}}$ and the $g_{a\mathrm{BB}}$ couplings are quantized in the units of $eg$ and $g^2$, respectively. Moreover, in the case of the trivial topology of the base manifold, there is no a priori reason for the latter three couplings to be quantized. Second, contrary to what has been advocated recently in the literature, we found that magnetic monopoles of an Abelian gauge field need not give mass to axions coupled to this gauge field. We showed that the axions do get mass in theories with magnetic monopoles only if these monopoles carry an additional instanton degree of freedom interacting with the axion $a$, e.g. in the case of the spontaneously broken symmetry phase of a non-Abelian gauge theory with $aGG^d$ term in the Lagrangian, where $G$ ($G^d$) is the (dual) field strength tensor of the non-Abelian gauge field. This axion mass is generated via the conventional mechanism through instantons of the non-Abelian theory, which however live on the 't Hooft-Polyakov monopoles in the low energy phase. Finally, we found that the interaction between axions and an electromagnetic field need not preserve CP. In particular, as long as one has heavy dyons in the high energy theory, there exists a natural source of CP-violation which can be transferred to low energy physics through axion-photon interactions. We discussed experiments which can probe this new CP-violating axion coupling.

### Acknowlegments

We thank P. Quilez for bringing the argument about the quantization of the axion-photon coupling to our attention as well as for numerous valuable discussions at the beginning stages of this project. A.R. acknowledges support and A.S. is funded by the Deutsche Forschungsgemeinschaft (DFG, German Research Foundation) under Germany's Excellence Strategy – EXC 2121 *Quantum Universe* – 390833306. This work has been partially funded by the Deutsche Forschungsgemeinschaft (DFG, German Research Foundation) - 491245950.

## A  Calculation of the anomaly coefficients

Let us calculate the anomaly coefficients $E$, $M$ and $D$, which enter the expressions (5.3), (5.4) and (5.6) for the axion-photon couplings. We start with a high energy QEMD theory and integrate out heavy fermions $\psi$ carrying PQ charges. In the PQ-symmetric phase, the Lagrangian includes the following terms for each of the fermions $\psi$:

$$\mathcal{L} \supset i\bar{\psi}\gamma^\mu \mathcal{D}_\mu \psi + y \left( \Phi\, \bar{\psi}_L \psi_R + \mathrm{h.c.} \right) , \tag{A.1}$$

where $y$ is a dimensionless Yukawa constant, $\mathcal{D}_\mu = \partial_\mu - e\, q_\psi A_\mu - g_0\, g_\psi B_\mu$ is a covariant derivative including both magnetic and electric four-potentials multiplied by the corresponding electric and magnetic charges, and $\Phi$ is the PQ complex scalar field.

Below the PQ symmetry breaking scale, one can expand $\Phi = (v_a + \rho) \exp(ia/v_a)/\sqrt{2}$, where $\rho$ is a heavy radial mode and $a$ is an axion. The terms in the resulting Lagrangian

which are relevant for the low energy phenomenology are:

$$\mathcal{L} \supset i\bar{\psi}\gamma^\mu \mathcal{D}_\mu \psi + \frac{yv_a}{\sqrt{2}}\left\{\exp\left(\frac{ia}{v_a}\right)\bar{\psi}_L\psi_R + \text{h.c.}\right\}. \tag{A.2}$$

We perform an axial rotation of the fermion $\psi \to \exp\left(ia\gamma_5/2v_a\right)\cdot\psi$, after which there arise an anomalous contribution $\mathcal{L}_\text{F}$ from the transformation of the measure of the path integral and a derivative coupling of $a$ to the axial current of $\psi$:

$$\mathcal{L} \supset i\bar{\psi}\gamma^\mu \mathcal{D}_\mu \psi + \frac{yv_a}{\sqrt{2}}\bar{\psi}\psi - \frac{\partial^\mu a}{2v_a}\bar{\psi}\gamma_\mu\gamma_5\psi - \mathcal{L}_\text{F}. \tag{A.3}$$

The fermion $\psi$ gets its mass $m = yv_a/\sqrt{2}$, which we assume to be large compared to the energy scales probed in experiments. In the large $m$ limit, the derivative axion coupling from Eq. (A.3) does not contribute to the low energy Lagrangian of axion-photon interactions due to the Sutherland-Veltman theorem [135, 136]. The axion-photon couplings are thus given by the anomalous terms $\mathcal{L}_\text{F}$ which can be calculated using the Fujikawa method [137]. In particular, following Fujikawa, we apply the gauge invariant heat kernel regularization to the path integral measure which yields:

$$\mathcal{L}_\text{F} = -\frac{a}{v_a} \cdot \lim_{\substack{\Lambda\to\infty \\ x\to y}} \text{tr}\left\{\gamma_5 \exp\left(\slashed{\mathcal{D}}^2/\Lambda^2\right)\delta^4(x-y)\right\}. \tag{A.4}$$

The expression for $\slashed{\mathcal{D}}^2$ is:

$$\slashed{\mathcal{D}}^2 = \mathcal{D}^2 - i\gamma_\mu\gamma_\nu\left(eq_\psi\,\partial^\mu A^\nu + g_0 g_\psi\,\partial^\mu B^\nu\right). \tag{A.5}$$

It is then convenient to express the delta-function as a superposition of plane waves: $\delta^4(x-y) = \int d^4k\, e^{ik(x-y)}/(2\pi)^4$, each of which shifts the derivative operator $\mathcal{D}_\mu \to \mathcal{D}_\mu + ik_\mu$. Taking into account Eq. (A.5), we obtain:

$$\mathcal{L}_\text{F} = -\frac{a}{v_a} \cdot \lim_{\Lambda\to\infty} \int \frac{d^4k}{(2\pi)^4}\,\text{tr}\left\{\gamma_5 \exp\left(-i\gamma_\mu\gamma_\nu\left(eq_\psi\,\partial^\mu A^\nu + g_0 g_\psi\,\partial^\mu B^\nu\right)/\Lambda^2 + \right.\right.$$
$$\left.\left. (\mathcal{D}+ik)^2/\Lambda^2\right)\right\}. \tag{A.6}$$

Any terms in the integrand which are $o\left(1/\Lambda^4\right)$ vanish after performing the integration and sending $\Lambda \to \infty$. Taylor expanding the exponent, we are then left with the finite number of terms, which after taking the trace, the integral and the limit simplify into:

$$\mathcal{L}_\text{F} = \frac{a\,d(C_\psi)}{8\pi^2 v_a} \cdot \epsilon_{\mu\nu\lambda\rho}\left(eq_\psi\,\partial^\mu A^\nu + g_0 g_\psi\,\partial^\mu B^\nu\right)\left(eq_\psi\,\partial^\lambda A^\rho + g_0 g_\psi\,\partial^\lambda B^\rho\right), \tag{A.7}$$

where $d(C_\psi)$ is the dimension of the color representation of $\psi$. Using the notations of sec. 4, Eq. (A.7) can be rewritten as follows:

$$\mathcal{L}_\text{F} = -\frac{a\,d(C_\psi)}{16\pi^2 v_a} \cdot \left(q_\psi^2 e^2\,\text{tr}\left\{(\partial\wedge A)\,(\partial\wedge A)^d\right\} + g_\psi^2 g_0^2\,\text{tr}\left\{(\partial\wedge B)\,(\partial\wedge B)^d\right\} + \right.$$
$$\left. 2\,q_\psi g_\psi\,eg_0\,\text{tr}\left\{(\partial\wedge A)\,(\partial\wedge B)^d\right\}\right). \tag{A.8}$$

If there are multiple dyons $\psi$, each of them gives a similar contribution to the resulting axion-photon Lagrangian. Thus, the expressions for the axion-photon couplings $g_{a\mathrm{AA}}, g_{a\mathrm{BB}}, g_{a\mathrm{AB}}$ and the corresponding anomaly coefficients $E, M, D$ are:

$$g_{a\mathrm{AA}} = \frac{Ee^2}{4\pi^2 v_a}\,, \quad E = \sum_\psi q_\psi^2 \cdot d(C_\psi)\,, \tag{A.9}$$

$$g_{a\mathrm{BB}} = \frac{Mg_0^2}{4\pi^2 v_a}\,, \quad M = \sum_\psi g_\psi^2 \cdot d(C_\psi)\,, \tag{A.10}$$

$$g_{a\mathrm{AB}} = \frac{Deg_0}{4\pi^2 v_a}\,, \quad D = \sum_\psi q_\psi g_\psi \cdot d(C_\psi)\,. \tag{A.11}$$

In Ref. [81], we obtained the same results for the axion-photon couplings (A.9)-(A.11) directly integrating over the fermion fields in the path integral of the theory and discussing all the subtleties associated to the non-locality and $n_\mu$-dependence in detail. The associated calculation can be considered as a proof of the validity of the Fujikawa method presented in this Appendix.

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
