# Peer review of "Electromagnetic Couplings of Axions"

_SciPost Physics_

## Round 2 · Referee Report · Matthew Reece (Referee 1) · 2023-7-4

Strengths

The paper raises interesting questions about how axion physics relates to electric-magnetic duality. It is thought-provoking. It motivated me to think through interesting questions that I had not properly thought about before.

Weaknesses

The paper incorrectly claims that the Witten effect arises only in specific UV completions of axion electrodynamics. This mistake leads the authors to conclude that a variety of new axion-photon couplings are possible. While these new couplings are possible in abstract axion electrodynamics, they are unfortunately not viable in the Standard Model, because the Witten effect would then imply that Standard Model charged particles have magnetically charged, dyonic excitations. The underlying misunderstanding of the Witten effect invalidates the conclusions of the paper.

Report

This paper is thought-provoking. The central question is: in light of electric-magnetic duality, why do we always find axion couplings to photons proportional to $e^2$? Electric-magnetic duality exchanges $e$ with $2\pi/e$, so shouldn't there be theories with much larger couplings proportional to $1/e^2$? The answer to this question is not obvious (at least to many particle theorists). The paper makes very interesting claims about new axion-photon couplings, which fit with the intuition that electric-magnetic duality should allow much stronger couplings. From that viewpoint, I am happy that it appeared on arXiv. It caused me to think more carefully about things I had not properly thought through before. Based on many conversations I have had in the last year, a number of other physicists interested in axions had also not thought them through. Unfortunately, the paper reaches incorrect conclusions.

Let me make a few remarks, for the sake of full disclosure: I first contacted the authors of this paper in May of 2022, after their initial preprint appeared on arXiv, to explain to them that I thought its conclusion was incorrect. They did not agree with my argument. I subsequently served as an anonymous referee when this paper was submitted to JHEP, in September of last year. I wrote a detailed referee report explaining why the paper's conclusions are incorrect. Some of my understanding of this owes to conversations with Ben Heidenreich and Jake McNamara, with whom I have written a paper draft explaining the following claims in more detail, which we will post to arXiv in the near future.

After our extensive previous communication, and presumably also discussions with other physicists, the authors have written a second version. It is now very clear that the main source of our disagreement lies in the validity of the Witten effect. Thus, I will not rehash all the comments I made in my previous report for JHEP, but will focus on explaining the Witten effect. The authors claim that the Witten effect arises only in particular UV completions of axion electrodynamics. This is not true. The Witten effect is a generic effect that arises in any theory with a $\theta F \wedge F$ coupling. It is essentially a form of anomaly inflow. This claim has been well-known for several decades, and one can find more or less equivalent statements in classic 1980s papers and reviews by Coleman, Preskill, Wilczek, and others. The Witten effect is now a completely accepted and standard aspect of quantum field theory.

Here is the clearest quick derivation I know for the basic physics behind the Witten effect in the axion context. Consider free Maxwell theory without a theta term, with gauge field $A$ and field strength $F = dA$. Let us work away from any point-like electric or magnetic sources. To introduce a magnetic gauge field $B$, one seeks to solve the equation $\frac{1}{2\pi} dB = \frac{1}{e^2} \star F$. The left-hand side is the magnetic flux of $B$ and the right-hand side is the electric flux of $A$. They should be interchanged by electric-magnetic duality. Can this equation be solved for $B$? Locally, yes, it always can. The reason is that Maxwell's equations tell us that (away from charged particles or currents) $d\star F = 0$. This means that locally, $\star F$ is a derivative, and we introduce $B$ as the quantity it is a derivative of. (This is just a local statement; globally, we might have to patch together different choices of $B$ on different charts, but if $B$ is a solution so is $B + d\lambda$, the basic gauge invariance that allows us to do such patching.)

In axion electrodynamics, even away from sources, we do not have $d \star F = 0$. Instead, we have $d\star F = \frac{e^2}{4\pi^2} d\theta \wedge F$. So we can't introduce $dB$ in the same way! It just doesn't make sense. We have to modify the definition. One modification is to note that (again, away from sources) $d(\frac{1}{e^2} \star F - \frac{1}{4\pi^2} \theta F) = 0$. So we can define the magnetic gauge field to be the quantity whose derivative is the expression in parentheses, $\frac{1}{2\pi} dB = \frac{1}{e^2} \star F - \frac{1}{4\pi^2} \theta F$. This makes sense; there indeed (locally) exists a solution for $B$. But now $B$ is not invariant when $\theta \to \theta + 2\pi$. Instead, $B \to B - A$. This is the Witten effect!

This derivation -- which readers familiar with $SL(2,\mathbb{Z})$ duality will recognize as the shift in the magnetic gauge field under the $T$ element of the duality group -- shows that the derivation of the Witten effect is completely independent of details of strong fields near singular sources. A magnetic monopole, by definition, should couple to $B$. But $B$ is not gauge invariant when we shift the axion by $2\pi$, so the coupling must be modified. This is why monopoles must have a dyonic mode in axion electrodynamics. However, I would emphasize that my derivation above did not involve monopoles at all. Indeed, the action of $SL(2,\mathbb{Z})$ on electric and magnetic gauge fields is something one can derive in free Maxwell theory by integrating Lagrange multiplier fields into the path integral and then integrating out the original fields (see, e.g., Witten's classic https://arxiv.org/abs/hep-th/9505186).

One can implement a variety of non-standard axion couplings by exploiting electric-magnetic duality, which is essentially what the authors have done. But then, adapting the arguments above, we learn that electrically charged particles become dyons under $\theta \to \theta + 2\pi$. And this is a problem, because the Standard Model electrically charged fermions are chiral; they get mass only from electroweak symmetry breaking. For them to have dyonic cousins that they turn into when $\theta$ varies, we must couple the $\theta$ term to their mass in some way, and this means the dyons also must be chiral. This is completely inconsistent with Higgs physics. (A more detailed argument will appear in my paper on arXiv soon.)

As I said above, I think the paper is thought-provoking. It's unfortunate that the effects it proposes simply do not make sense when coupled to the Standard Model. The authors and others have written several follow-up papers, including papers proposing new experiments. Unfortunately, these experiments would be searching for effects that cannot exist in consistent quantum field theory. I think that it is important that experimentalists can find a correct treatment of these questions, before undertaking such a search.

Requested changes

There are many interesting ideas in the paper, but the conclusions should not remain in anything resembling their current form.

  • validity: poor
  • significance: high
  • originality: high
  • clarity: good
  • formatting: excellent
  • grammar: excellent

Author:  Anton Sokolov  on 2023-07-27  [id 3844]

(in reply to Report 1 by Matthew Reece on 2023-07-04)
Category:
reply to objection

We thank Prof. Reece for writing a referee report on our manuscript submitted to SciPost. In his report, Prof. Reece mentions that he previously refereed our manuscript for JHEP. We have to note that unfortunately, JHEP did not allow us to reply to this previous report. We are glad that we can now reply to the concerns raised by Prof. Reece here. In the previous report, there were two main objections from the referee: one against the Zwanziger theory and another against our treatment of the Witten-effect induced axion coupling. As the referee does not mention his concerns about the Zwanziger theory in the current report, we assume that there is no disagreement on this point anymore.

The remaining objection from Prof. Reece, which he explained in the current report, concerns the Witten-effect induced axion coupling $g_{aAj}$, which is a triple-vertex coupling between axion, magnetic monopole and electromagnetic field. In particular, he clearly contests our statement that the Witten-effect induced axion coupling $g_{aAj}$ and the standard axion-photon coupling $g_{a\gamma \gamma}$ are in principle two different couplings, i.e. that they can have different values $g_{aAj} \neq g_{a\gamma \gamma}$, see our detailed discussion of this issue in secs. (3.2)-(3.4) in the manuscript.

First of all, we would like to stress that Prof. Reece did not indicate any mistakes in the calculations and analysis given in our paper. The referee's claim is only that our final results in secs. (3.2)-(3.4) contradict some kind of a no-go theorem known to the referee, which states that the equality $g_{aAj} = g_{a\gamma \gamma}$ cannot be violated. In this reply, we will try to persuade the referee and the editor that such no-go theorem actually does not exist, therefore strengthening our arguments from secs. (3.2)-(3.4). For this, we will first explain why the derivation in the paragraphs 4-5 of the referee's report is unsatisfactory and thus cannot be used to disprove our work. Then, we will formulate the core of our disagreement with Prof. Reece in a strict mathematical language and make an exact calculation supporting our point of view. Finally, and probably most importantly, we will try to explain why the misinterpretation of our results by Prof. Reece arose in the first place, discussing the subtle points related to the $SL(2, \mathbb{Z})$ electric-magnetic duality in Abelian gauge theories and resolving the seeming paradox associated to the relevance/irrelevance of the $\theta$-term in these theories formulated in physical space-time.

1. Analysis of the argument given by Prof. Reece to disprove the results of our manuscript.

As we already mentioned, we find the derivation given by Prof. Reece in the paragraphs 4-5 of his report unsatisfactory. We oppose the following line of thought by the referee:

Consider free Maxwell theory without a theta term, with gauge field $A$ and field strength $F=dA$. Let us work away from any point-like electric or magnetic sources. To introduce a magnetic gauge field $B$, one seeks to solve the equation $\frac{1}{2\pi} dB=\frac{1}{e^2} \star F$. ... In axion electrodynamics, even away from sources, we do not have $d \star F = 0$. Instead, we have $d \star F=\frac{e^2}{4\pi^2}\, d\theta \wedge F$. So we can't introduce $dB$ in the same way! It just doesn't make sense. We have to modify the definition.

It is not true that one cannot introduce the dual four-potential $B$ in the axion electrodynamics in the same way as it is done in the free Maxwell theory. In fact, from the point of QFT, one has to introduce it in the same way. Axions and electrons are both particles represented by quantum fields, which both interact (albeit in different ways) with the electromagnetic field. The same way as the electromagnetic field can be considered away from electrons, it can be considered away from axions. Moreover, to define the degrees of freedom pertinent to the electromagnetic field itself, one has to consider it away from all the sources, including axions. The construction of the amplitude for any physical process starts by identifying the asymptotic states, which evolve according to the free theory, and so it does not make sense to choose the definition for the (dual) photon field $B$ such that it involves the interaction with axion $\theta$ by default. For any physical process involving axions and photons, in the asymptotically distant regions of spacetime both the photons and the axions propagate freely.

Let us further continue with the analysis of the derivation given by Prof. Reece:

One modification is to note that (again, away from sources) $d \left( \frac{1}{e^2} \star F - \frac{1}{4\pi^2} \theta F \right) = 0$. So we can define the magnetic gauge field to be the quantity whose derivative is the expression in parentheses, $\frac{1}{2\pi} dB = \frac{1}{e^2} \star F - \frac{1}{4\pi^2} \theta F$. This makes sense; there indeed (locally) exists a solution for $B$. But now $B$ is not invariant when $\theta \rightarrow \theta + 2\pi$. Instead, $B \rightarrow B-A$. This is the Witten effect!

We disagree also with this piece of derivation given by the referee. First of all, we have again the concerns that we outlined while commenting on the previous part of this derivation, i.e. that the dual photon $B$-field should not depend on the axion field $\theta$ by definition, since one should be able to separate the physical degrees of freedom pertinent to axions from the ones pertinent to electromagnetic field. Second, even if we impose the expression for $B$ suggested by the referee, i.e. we define some new field $B$ carrying both the degrees of freedom of the electromagnetic field and of the axions, one has to actually demonstrate how exactly these degrees of freedom fit into the four-vector field $B$, i.e. practically construct a viable kinetic term for the $B$-field, then construct the Hamiltonian for the theory and the associated Hilbert space. For example, if one chooses a usual Lagrangian for $B$, like for any massless four-vector gauge field (as it should be for the dual photon), then $B$ carries only two physical degrees of freedom of the electromagnetic field, and so the third degree of freedom pertinent to the axion field simply does not fit. Thus even if one adopts the definition of $B$ by Prof. Reece, which is basically some kind of a non-linear redefinition of the physical fields, it is not at all clear whether it is possible to construct a consistent theory for such a complicated field $B$ (let alone that there is no physical content in such redefinition in the first place, as we emphasized before). In any case, such $B$ would no longer represent what one usually means by the dual photon field.

To conclude, we showed that the arguments about the "proper" definition of the dual potential $B$ by Prof. Reece are incorrect. The $B$-field should by definition represent only the degrees of freedom pertinent to the electromagnetic field, and therefore be defined as the connection of a principal $U(1)$-bundle with the curvature $\star F$, so that $\frac{1}{2\pi} dB = \frac{1}{e^2} \star F$. The $B$-field is then obviously fully invariant with respect to the discrete shift transformations of the axion field $\theta$. The Witten-effect induced interactions of axions $g_{aAj}$ need not be the part of axion electrodynamics.

Finally, let us note that the incorrect derivation given by Prof. Reece cannot be found in any previous literature (at least in well-known sources), which shows that the argument is not at all "standard", contrary to what the referee claims in the paragraph 3 of his report.

2. Core of the disagreement formulated mathematically and resolved robustly.

As Prof. Reece clearly stated, the main point of the disagreement can be formulated in even more simple terms if one substitutes the axion field by the constant $\theta$: we basically disagree on whether the $\theta$-term ($\theta F \wedge F$) in the Lagrangian of the $U(1)$ gauge theory necessarily leads to the Witten effect. To resolve the disagreement, let us do a robust mathematical analysis of the $\theta$-term in $U(1)$ gauge theory.

Mathematically, $U(1)$ gauge theory in space-time $M$ is described by a principal $U(1)$-bundle $\pi : P \rightarrow M$. The curvature 2-form $F$ of the latter principal bundle does not depend on the choice of the principal connection 1-form $iA$ valued in the Lie algebra $\mathfrak{u}(1) \cong i\mathbb{R}$. Moreover, due to the Abelian nature of the structure group, the relation between these two forms is particularly simple: $F=dA$. Given that the curvature $F \in \Omega^2(M)$ is $U(1)$-invariant, the action functional for the theory is defined on the space of all the connections $\mathcal{A}(P)$ as follows:

$$ S [ A ] = \int_M \left( -\frac{1}{2e^2}\, F \wedge \star F \; + \; \frac{\theta}{4\pi^2}\, F\wedge F \right) \, . \qquad (1) $$
If one assumes $M=T\times \mathbb{R}^3$, which is topologically equivalent to the physical case of flat space-time, then the considered principal bundle is trivial, and no charge quantization arises.

However, we are interested in studying the $U(1)$ gauge theory in the presence of a magnetic monopole, which yields a completely different scenario: in this case, the topology of $M$ has to be modified. Indeed, suppose there exists a static magnetic monopole at the origin of the spatial coordinates $\lbrace 0 \rbrace \in \mathbb{R}^3$, and consider any sphere $S^2_0$ with the centre at $\lbrace 0 \rbrace$. By definition of the monopole, the magnetic flux $\int_{S^2_0} F \neq 0$, which due to the identity $F=dA$ means that there cannot exist a global projection of the principal connection $A$ onto the sphere $S^2_0$ and therefore onto the base space $M$. The obstruction to the triviality of the principal bundle is described by the relevant characteristic class, which for the principal $U(1)$-bundle is the first Chern class $c_1 \in H^2(M, \mathbb{Z})$ valued in the second cohomology group of $M$ with integer coefficients. Since the corresponding Chern form, i.e. the invariant polynomial is simply $-F/2\pi$, and $H^2(S^2_0, \mathbb{Z}) \cong \mathbb{Z}$, the magnetic flux is quantized: $\int_{S^2_0} F = 2\pi n$, $n \in \mathbb{Z}$, which of course yields the Dirac quantization condition for the charges. Now, we see exactly why the topology of $M$ should be modified: the second cohomology group is always non-trivial in the presence of a magnetic charge, and therefore our space $\mathbb{R}^3$ actually has a hole: $M = T\times \mathbb{R}^3 \backslash \lbrace 0 \rbrace$.

Note that due to the hole in the base space $M$, it is meaningless to assign any value to $F$ at $\lbrace 0 \rbrace$: this point is simply excluded from our space. For example, within the $U(1)$ gauge theory reviewed in the previous paragraphs, it does not make sense to write $\varepsilon^{0\mu \nu \lambda}\partial_{\mu} F_{\nu \lambda} = 4\pi \delta^3 (\mathbf{x})$: the support of the distribution falls exactly into the hole invalidating the statement. Of course, one could choose a different, more complete theoretical framework, such as Zwanziger's (Schwinger's) quantum relativistic theory of magnetic charges, where no such holes arise; however, in the latter theories, one has to introduce the magnetic charges directly into the Lagrangian, as well as to modify the structure of the $U(1)$ gauge theory significantly, as it was illustrated in sec. 4 by the example of the Zwanziger theory. In our manuscript, we give a description of the Witten effect (as well as the Rubakov-Callan effect) within the Zwanziger theory.

Now, staying in the framework of the $U(1)$ gauge theory described earlier, let us derive the variation of the action (1) in the presence of a magnetic monopole and show that there arises no contribution from $\theta$, i.e. that no electric charge for the monopole is generated by the $\theta$-term. First, we decompose the electromagnetic field into the dynamical field $F_l$ and the background field $F_b$: $F = F_l + F_b$, where $F_b$ is the field of the monopole. As the monopole is an external source, we have to vary only the four-potential $A_l$ ($F_l = dA_l$), which is defined globally everywhere in $M = T\times \mathbb{R}^3 \backslash \lbrace 0 \rbrace$, because there are no dynamical magnetic charges by assumption. Second, we describe the boundary of the hole in $M$ by an infinitesimal sphere $S^2_{\varepsilon}$ with radius $r \to 0$, and the boundary of $M$ at infinity as a sphere $S^2_{\infty}$ with radius $r \to \infty$, as usual. The variation of the action (1) is:

$$ \delta S \; = \; -\frac{1}{e^2} \int_M d \star F \wedge \delta A_l \; + \; \int_T \left( \int_{S^2_{\varepsilon}} K + \int_{S^2_{\infty}} K \right) \, , \quad K = \left( -\, \star F/e^2 + \theta F / 2\pi^2 \right) \wedge \delta A_l \, , \qquad (2) $$
where we used the Bianchi identity $dF = 0$ which is valid everywhere in $M = T\times \mathbb{R}^3 \backslash \lbrace 0 \rbrace$ (since $F$ is an invariant polynomial corresponding to the principal $U(1)$-bundle we consider, one can also view $dF = 0$ as one of the statements of the renowned Chern-Weyl theorem, although quite trivial in this case due to $d^2 = 0$). We also omitted the total time derivatives in the integrand. Finally, we formally put the variations at the boundaries to zero: $\delta A_l \vert_{S^2_{\infty}} = \delta A_l \vert_{S^2_{\varepsilon}} = 0$, i.e. we assume that the boundary integrals in the variation of the action (2) vanish (see a detailed justification further in the text). The resulting Euler-Lagrange equations are:
$$ d \star F = 0 \, , \qquad (3) $$
so we see that there is no extra electric current induced by the $\theta$-term.

For the discussion to be exhaustive, let us analyze whether there could arise any non-trivial effects from the surface terms in the variation of the action (2). Although within the Euler-Lagrange method, we put the variations of fields at the boundaries to zero, it is known that in non-Abelian gauge theories, semi-classical (finite-action) IR processes are possible which yield $\delta A_n \vert_{S^2_{\infty}} \neq 0$ (non-zero, but a pure gauge), where $A_n$ is the non-Abelian four-potential. These processes called instantons can modify the IR dynamics of the non-Abelian theory. However, no similar effects arise in the $U(1)$ gauge theory case under consideration. Indeed, due to the finiteness of the action, the field strength $F$ has to fall faster than $\rho^{-2}$ at infinity, while the four-potential falls as $\delta A_l = O(\rho^{-1})$, where $\rho = \sqrt{x_{\mu} x_{\mu}}$. This means that the integral of $K$ over the infinity 3-surface $\Sigma_{\infty}$ equals zero in the limit $\rho \to \infty$: no instanton solution exists. Also, this is clear from the topology of the $U(1)$ group, since $\pi_3 (U(1)) = 0$, and no non-trivial winding is possible.

It remains to analyze the boundary term at $S^2_{\varepsilon}$. It may seem confusing that the condition $\delta A_l \vert_{S^2_{\varepsilon}} = 0$, imposed by the Euler-Lagrange method, has to be satisfied: why should the value of the four-potential be fixed in the vicinity of $\lbrace 0 \rbrace$? Actually, we will show now that the $S^2_{\varepsilon}$ surface term in the variation of the action vanishes regardless of whether we impose this condition. First, let us note that for an infinitesimal sphere, $A_l$ must be the same at every point of it: we can always take the radius $r_{\varepsilon}$ of the sphere to be much smaller than the spatial variation of the four-potential. If the monopole has structure originating from some UV physics, then this is simply a statement that the field $A_l$ is a low energy field by the definition of the IR theory, and therefore its spatial variation is negligible compared to the UV short-distance scale. A homogeneous vector field integrated over $S^2_{\varepsilon}$ gives zero, so the only non-zero contribution to the surface integral could come from the terms in the integrand containing the background $F_b$ field:

$$ \int_T \int_{S^2_{\varepsilon}} K \; = \; \int_T \int_{S^2_{\varepsilon}} \left( -\, \star F_b/e^2 + \theta F_b / 2\pi^2 \right) \wedge \delta A_l \; = \; \frac{\theta}{2\pi^2} \int_T dt\, \delta \phi_l \vert_{S^2_{\varepsilon}} \int_{S^2_{\varepsilon}} d\mathbf{S}_{\varepsilon} \cdot \mathbf{B}_b \; , \qquad (4) $$
where $\delta \phi_l \vert_{S^2_{\varepsilon}}$ is the variation of the scalar potential on $S^2_{\varepsilon}$, $B_b$ is the background magnetic field, and $dS_{\varepsilon}$ is an oriented element of $S^2_{\varepsilon}$. The surface integral at the end of the Eq. (4) is simply the magnetic charge of the monopole, which is time-independent by construction, so we conclude that the pure gauge variations $\delta \phi_l \vert_{S^2_{\varepsilon}} = \dot{\alpha}$ change the Lagrangian of the system by a total time derivative and therefore are irrelevant. This means we can always set $\delta \phi_l \vert_{S^2_{\varepsilon}}$ to zero by performing a suitable gauge transformation at the boundary sphere $S^2_{\varepsilon}$. The surface integral (4) has thus no relevance for the dynamics of the system.

To conclude, we showed that the total variation of the action (1) does not depend on $\theta$, including any possible surface terms. This means that the saddle point of the path integral of the theory also has no $\theta$-dependence, and so the $\theta$-term has no influence on the physical processes at least at leading order. This means that there does not exist any no-go theorem saying that in the presence of the $\theta$-term in a $U(1)$ gauge theory, there cannot exist magnetic monopoles without extra electric charge of $\theta e/2\pi$. Whereas we presented a detailed analysis here, this conclusion could actually be easily guessed from the very beginning. Indeed, within the $U(1)$ gauge theory presented earlier, a magnetic monopole is an external source and one only studies the electromagnetic theory in its background. It is not surprising that one cannot change the properties of the external source, such as its electric charge, by adding extra terms into the Lagrangian: in the end, by definition of the external source all its degrees of freedom are independent of the dynamics of the theory and therefore of the Lagrangian. The Witten effect arising for example in the case of the 't Hooft-Polyakov monopoles can of course easily be accounted for in this theory: one should give the magnetic monopole an electric charge proportional to $\theta e/2\pi$ by definition. If the mass of such monopole (i.e. unification scale) is high enough, no low energy dynamics can change the value of this charge as well as its motion. If one wants to incorporate the monopoles into the theory as dynamical sources, then one has to use a more complete theoretical framework, as it was done in our manuscript. Namely, the implementation of the Witten effect as well as the Rubakov-Callan effect into such more complete theoretical framework, where the monopoles are allowed to be dynamical, was discussed by us in the manuscript.

3. Why the results of our work are fully consistent with the electric-magnetic $SL(2,\mathbb{Z})$ duality.

After disproving the argument given by Prof. Reece and after analyzing the core of the disagreement via robust mathematical methods, we proceed to explain the reason for the misinterpretation of our results by the referee. The original source of the confusion can be found by looking at the paragraph 6 of the report by Prof. Reece, where he mentions the concept of the $SL(2,\mathbb{Z})$ duality several times. Basically, the concern by the referee is that our results contradict what is known about the $SL(2,\mathbb{Z})$ duality. Indeed, in the framework of the $SL(2, \mathbb{Z})$ duality, it is well-known that the "$\theta$-term" ($ \theta \mathcal{F} \wedge \mathcal{F}$) plays an important role, and in particular determines the electric charges of dyons, i.e. represents the Witten effect. According to the referee, this latter statement contradicts our results, since in our manuscript (as well as in sec. 2 of this answer) we argue that the $\theta$-term ($ \theta {F} \wedge {F}$, see sec. 2 of this answer) does not represent the Witten effect.

As the reader could have already guessed, the misunderstanding arises due to the incorrect identification $\mathcal{F} = F$ by Prof. Reece. Indeed, if the field strength tensor $F = dA$ is used in the $\theta$-term, then, as we proved in sec. 2 of this answer, there is no Witten effect associated to the $\theta$-term. Now, to understand in what way $\mathcal{F}$ is actually different from $F$, it is enough to look into the seminal papers on the electric-magnetic $SL(2,\mathbb{Z})$ duality. The referee mentions the paper by Witten (https://arxiv.org/abs/hep-th/9505186), however Witten himself indicates right in the first sentence of the abstract that the $SL(2,\mathbb{Z})$ duality is a well-known result obtained before; the actual seminal papers which the Witten's paper refers to are the classic Cardy's and Rabinovici's papers (https://doi.org/10.1016/0550-3213(82)90463-1, https://doi.org/10.1016/0550-3213(82)90464-3). In particular, Cardy's paper (https://doi.org/10.1016/0550-3213(82)90464-3) establishes for the first time the full $SL(2,\mathbb{Z})$ duality of Abelian gauge theories with both electric and magnetic charges. The Lagrangian used by Cardy has the form similar to Eq. (1) of this answer, but it is crucial that instead of the field strength tensor $F=dA$, Cardy introduces the tensor $\mathcal{F} = F - 2\pi s$, which contains an extra contribution $-2\pi s$ associated to the magnetic current 3-form $m$: the 2-form $s$ is essentially introduced via the definition $d \star s = m$. Therefore, the tensor $\mathcal{F}$, contrary to the field strength tensor $F$, contains both the degrees of freedom pertinent to the electromagnetic field and the degrees of freedom pertinent to the magnetic charges. While it is possible to express the Lagrangian solely in terms of $\mathcal{F}$ when discussing the $SL(2,\mathbb{Z})$ duality of the theory of the interactions between electric and magnetic charges, any newly introduced particle such as axion need not interact with both the electromagnetic field and the magnetic charges in such a way that the interaction Lagrangian can be written solely in terms of this "composite" object $\mathcal{F}$. Axion $a$ can well have only the coupling to the electromagnetic field $F$ of the form $a\, F\wedge F$ ($g_{aAA}$ in the manuscript), but not have the coupling to the magnetic monopoles $m$ of the form of the Witten-effect induced interaction ($g_{aAj}$ in the manuscript). The consistent implementation of both of these couplings within the theory with both electric and magnetic charges was done in the manuscript using the Zwanziger theory.

We hope it is now clear that the would-be no-go theorem $g_{aAA} = g_{aAj}$ claimed by the referee originates from the confusion on the separation of the degrees of freedom between monopoles and electromagnetic fields. Although we have already presented a lot of evidence (and from different viewpoints) that such no-go theorem does not exist, let us continue with one more argument. In fact, one of the best ways to disprove a no-go theorem is to construct an explicit counter-example. This is exactly what we did in a follow-up paper (https://doi.org/10.48550/arXiv.2303.10170), where using the path integral techniques, we derived the IR effective Lagrangian for a concrete KSVZ-like axion model and obtained $g_{aAA} \neq g_{aAj}$. We stress that in his report and in the correspondence with us, Prof. Reece has not indicated any mistakes neither in the calculations and analysis presented in the manuscript under consideration in SciPost, nor in the follow-up paper we have just mentioned.

To conclude, in this answer we thoroughly addressed the concern by Prof. Reece that the main results of our paper could contradict some kind of a no-go theorem known to the referee. We first explained why the derivation of this would-be theorem given in the referee report is unsatisfactory. Second, we formulated the core of the disagreement in a mathematically precise way and made an explicit calculation supporting the point of view presented by us in the manuscript. Finally, we identified the source of the misunderstanding by Prof. Reece, and explained why our results are fully consistent with the concept of the electric-magnetic $SL(2,\mathbb{Z})$ duality.

Anonymous on 2023-07-28  [id 3850]

(in reply to Anton Sokolov on 2023-07-27 [id 3844])

Dr. Sokolov has written a long reply; I will just briefly respond to two points.

First, Dr. Sokolov claims that 'the incorrect derivation given by Prof. Reece cannot be found in any previous literature (at least in well-known sources), which shows that the argument is not at all "standard", contrary to what the referee claims.'

The argument I have given is that, in the presence of a Chern-Simons term, electric-magnetic duality trades it for a non-standard Bianchi identity for the gauge field. This modifies the proper definition of a magnetic gauge field. Whether this appears in “well-known sources” may depend on one’s definition of “well-known”, but let me first mention that already in email correspondence over a year ago, I pointed Dr. Sokolov to

Don Marolf, "Chern-Simons terms and the Three Notions of Charge," https://arxiv.org/abs/hep-th/0006117

which discusses such physics (e.g., on page 3, "One can often exchange a modified Bianchi identity for a Chern-Simons term by performing an electromagnetic duality transformation.")

In the report I wrote for JHEP last year, I also pointed him to

Hajime Fukuda and Kazuya Yonekura, "Witten effect, anomaly inflow, and charge teleportation," https://arxiv.org/abs/2010.02221,

a modern reference that discusses exactly the issue under consideration here.

The general physics under which electric-magnetic duality interchanges Chern-Simons terms and modified Bianchi identities (or Stueckelberg couplings) is absolutely pervasive in literature on supergravity and string theory (e.g., in the context of RR fields and D-branes, or the Green-Schwarz mechanism). More references that may qualify as "well-known" are

Edward Witten, "Some Properties of O(32) Superstrings," https://inspirehep.net/literature/15631,

which pointed out that axions arise in heterotic string compactifications from the B-field. The 2-form B-field has a modified Bianchi identity (Green-Schwarz) but can be dualized to a 4d axion with a Chern-Simons coupling. This is an exactly parallel kind of modification to electric-magnetic duality to what we are discussing. Some textbook references that refer to the connection between modified Bianchi identities and Chern-Simons terms are

Joseph Polchinski, String Theory vol. 2, chapter 12
Alessandro Tomasiello, Geometry of String Theory Compactifications, chapter 1 (see the discussion of "twisted field strengths" for RR fields)

Searching for the phrase "modified Bianchi identity" may be the easiest way to find a large number of sources explaining this physics. Whether it is "well-known" or not may be a matter of definition or of the community of researchers that one interacts with. In any case, there is a large body of literature on this topic, and I would suggest that any physicist who wants to study electric-magnetic duality in the presence of axions should acquaint themselves with it.

Second, Dr. Sokolov’s counter-argument to the Witten effect involves imposing a Dirichlet boundary condition that sets variations of the gauge potential to zero at the location of the monopole core. The argument that I gave implies that the worldline coupling of a monopole to the magnetic gauge field B is not gauge-invariant under shifts of the axion. The usual way to repair this is by anomaly inflow, implying the existence of a dyon mode. Instead, one might try to evade this argument by imposing a Dirichlet boundary condition on either the axion or the electric gauge field along the monopole worldline. I think that this may be a valid procedure for defining static probe monopoles (i.e., ’t Hooft lines). However, for dynamical monopoles, such boundary conditions are unphysical. They are limiting cases of the standard monopole action with a dyonic mode, where the coefficient of the dyon kinetic term is either sent to zero (leading to a Dirichlet boundary condition on the axion) or to infinity (leading to a Dirichlet boundary condition on the gauge field). The former case implies that the axion is very strongly coupled to the monopole, and monopole loops give the axion an enormous mass. The latter case, by contrast, implies that the energy gap between the monopole and the dyon states goes to zero. Neither case has the physics that Dr. Sokolov would like to find.

Author:  Anton Sokolov  on 2023-08-01  [id 3864]

(in reply to Anonymous Comment on 2023-07-28 [id 3850])
Category:
reply to objection

In his reply to our answer, Prof. Reece raises two different points. First of them addresses our doubt on whether the argument given in the paragraphs 4-5 of the referee report can be found in well-known sources. While we appreciate the detailed response on this issue, we actually consider this concern of ours as the least significant part of our answer: we believe that the most scientifically important question is whether the argument by the referee is correct, but not whether it can be found elsewhere. As Prof. Reece did not provide any objections to our rebuttal of the argument in question, we assume he agrees with the analysis presented in sec. 1 of our answer, and so also with our conclusion that the argument from the paragraphs 4-5 of the referee report is unsatisfactory. The latter conclusion is the most significant point we wanted to convey in sec. 1 of our answer. The remaining disagreement on the question of sources is less significant, so let us return to it later.

1. Analysis of the second point raised by the referee

The second point raised by the referee is an objection against the argument given by us in sec. 2 of our answer. The referee claims that our derivation depends on an unjustified assumption, namely on the Dirichlet boundary condition for the gauge potential at the location of the monopole. We strongly disagree with the referee on this point. We stress that it is not true that we consider only the Dirichlet boundary condition for the gauge field at the location of the monopole. Indeed, in the second to last paragraph of sec. 2 of our answer, we write:

It remains to analyze the boundary term at $S^2_{\varepsilon}$. It may seem confusing that the condition $\delta A_l \vert_{S^2_{\varepsilon}} = 0$, imposed by the Euler-Lagrange method, has to be satisfied: why should the value of the four-potential be fixed in the vicinity of $\lbrace 0 \rbrace$? Actually, we will show now that the $S^2_{\varepsilon}$ surface term in the variation of the action vanishes regardless of whether we impose this condition.

Our subsequent analysis does not take advantage of the Dirichlet boundary condition; in fact, in the remaining part of this paragraph we indeed show that the surface term vanishes regardless of the boundary condition imposed on the variation of the gauge potential at the location of the monopole. Therefore, the objection by Prof. Reece to the argument in sec. 2 of our answer is invalid.

2. Analysis of the first point raised by the referee

Now, let us return to the first point in the reply by the referee, where he claims that the argument he gave in the paragraphs 4-5 of the report can be found in multiple sources. First of all, let us note that almost all of the sources (except one) mentioned by Prof. Reece investigate the Chern-Simons couplings specifically in the context of supergravity and/or string theory. However, the discussion in our manuscript, as well as the counter-argument given by the referee, is essentially a statement about a generic Abelian gauge theory. We do not object to the fact that the Witten-effect induced interactions of axions can exist, nor that in some models one obtains the equality between the axion-photon coupling $g_{a\gamma \gamma }$ and the Witten-effect induced coupling $g_{aAj}$. It is true that in the specific models previously considered in the literature, for example in the models of the string-theoretic axions, one obtained $g_{a\gamma \gamma } = g_{aAj}$. The anomaly inflow mechanism is crucial for these theories to retain their consistency, and so the Witten-effect induced interaction must exist. We by no means oppose this statement. However, it is one of the novel results of our work, discussed by us in secs. 3.2-3.4 and reinforced by our previous answer, that generically, Witten-effect induced interactions could be absent.

For example, consider the first source suggested by the referee: "Chern-Simons terms and the Three Notions of Charge" by Dr Don Marolf, https://arxiv.org/abs/hep-th/0006117. Using the terminology of this article, the presence of the Witten-effect induced interaction is equivalent to the Page charge, but not the brane charge, being quantized. The author argues that such quantization of the Page charge in supergravity actually follows from the quantization of the usual (brane) charge in the underlying 11-dimensional theory. He stresses in the end of sec. 4 that it is basically because of the Kaluza-Klein reduction of the 11-dimensional theory that the Page charge, but not the brane charge of the D4-branes in 10-dimensional theory is quantized: "Thus, it is the Page charge that lifts to the familiar notion of charge in 11-dimensions. Quantization of the usual charge in 11-dimensional Einstein-Maxwell theory directly implies quantization of D4-brane Page charge in ten-dimensions." We fully agree with this conclusion. What we are saying, and what is fully consistent with this conclusion, is that without the Kaluza-Klein reduction, in general, there is no a priori reason for the Page charge, but not the brane charge, to be quantized.

Finally, let us consider the one source from the referee's reply which addresses the general theory of axion electrodynamics, without limiting itself specifically to the framework of supergravity/string theory. This is the paper by Dr Hajime Fukuda and Dr Kazuya Yonekura "Witten effect, anomaly inflow, and charge teleportation," https://arxiv.org/abs/2010.02221. Let us note that this is a very recent paper, which means that the topic of general axion electrodynamics theory is currently an active area of research, and so we would not call the results presented in this paper "standard". We think that this paper provides a very clear explanation of the Witten-effect induced interaction of axions and especially its relation to the anomaly inflow mechanism, however we stress that the derivation of these effects given in sec. 2.1 is not generic, i.e. the authors do not prove that these effects arise in any possible theory of axion electrodynamics. Indeed, the authors of this article suggest one particular definition for the coupling of the gauge field to a magnetic brane in the presence of the Chern-Simons term, which has the form (see their Eq. (2.14)) similar to the definition of the Page charge from the previously discussed article, but they do not argue that this is the only possible way of defining this coupling. They discard the definition based on the form similar to the Maxwell charge ($\int_N \tilde{F}$, see their Eq. (2.10)), but do not claim that any other definition is impossible. For example, in our manuscript, we present the consistent definition of the coupling of the gauge field to a magnetically charged particle by introducing the dual gauge field via the Zwanziger theory. In this case, there is no need to define this coupling through neither of the two forms mentioned above. In fact, the definition in our manuscript turns out to be the most general one, since one does not have to introduce any arbitrary manifold of integration $N$ for defining the coupling (unlike in Eqs. (2.10) and (2.14) from the discussed article), and so one does not have to impose $N$-independence of the theory as an extra condition. While in the discussed article, this extra condition gives rise to the Witten-effect induced coupling, in our more general approach, the Witten-effect induced interactions need not be present, although of course they can arise if the dynamics of the underlying UV theory demands it, as it was discussed in the previous paragraph.

To conclude, we discussed in detail both points from the reply by Prof. Reece. First, we explained why his second point, which is an objection to the argument in sec. 2 of our previous answer, is invalid. Second, we addressed the remaining point by the referee, where he opposed the statement from the last paragraph of sec. 1 of our previous answer. In particular, we explained why the would-be no-go theorem presented by the referee in the paragraphs 4-5 of his report is not a standard result, while considering specifically the sources suggested by the referee in his reply. Finally, let us stress that the referee did not object to the main argument presented by us in sec. 1 of the previous answer, where we showed that this would-be no-go statement is incorrect.

---

## Round 2 · Referee Report · Anonymous (Referee 2) · 2023-7-18

Report

The article contains a lot of technicalities. However, I would like to comment on some potential fundamental inconsistencies at a conceptual level.

1) The effective theory, presented in section 2, claims that we cannot write an effective theory for the monopole. This is not a fully valid argument, since we can always write an effective theory at distances larger than the monopole size, a close relative of an effective low energy theory of a heavy point charge (e.g., heavy nuclei).

Instead, the authors try to circumvent this problem, using a description proposed by Zwanziger. In this theory, currents are charged with respect to both electric and magnetic charges. While Zwanziger's approach is fully sound for understanding certain properties of such theories, implementing it for reaching the current goals (in particular predicting the new electromagnetic coupling of axion at the accuracy of phenomenological precision) is less informative. The problem is that validity of this effective description is hard to verify, as the theory carrying the elementary particles with both types of charges (electric and magnetic) is expected to be strongly coupled and thereby out of control. For magnetically charged solitons in weakly coupled theory, such as 't Hooft-Polyakov monopoles, this is not an issue due to exponential suppression of their contribution into the virtual processes. But, for elementary particles with magnetic charges, the theory enters the strong coupling regime. In particular, the validity of classical equations of motions (e.g., motion of a point particle) cannot be trusted due to strong quantum back-reaction which is un-calculable. Consequently, the conclusions driven from such an effective description are unreliable.

2) Next, the authors consider the extra couplings with electromagnetism. From the point of view of UV-completion, such couplings are possible if the axion is shared between two Yang-Mills sectors (the authors acknowledge this on page 28 at the end of the second paragraph). In the case of the minimal QCD axion, this is not the case. On the other hand, if we introduce such gauge sectors by extending the standard model gauge symmetry and couple them to ordinary QCD axion, the solution to the strong CP problem is jeopardized, since the axion gets potential from two distinct sources, thereby no longer compensating the $\theta$-term of QCD. The actual mismatch is a quantitative issue, which can be secondary. However, the conceptual question is whether axion can acquire new couplings without offsetting its vacuum.

Because of the above major issues, I would recommend re-examination of the idea.

  • validity: -
  • significance: -
  • originality: -
  • clarity: -
  • formatting: -
  • grammar: -

Author:  Anton Sokolov  on 2023-09-07  [id 3961]

(in reply to Report 2 on 2023-07-18)
Category:
reply to objection

We strongly disagree with the claim by the referee that our manuscript contains any fundamental inconsistencies at a conceptual level. In fact, as we will show below, the issues raised by the referee have no relevance for our work. The arguments that the referee gives to disprove our work are therefore incorrect. Let us illustrate this claim of ours by addressing in detail both issues raised by the referee.

1. Analysis of the first issue raised by the referee

Objection 1

The first issue raised by the referee is organized into two objections corresponding to two paragraphs in the report. The first would-be inconsistency criticized by the referee is formulated within the report as follows:

The effective theory, presented in section 2, claims that we cannot write an effective theory for the monopole.

We disagree with this statement. In particular, we stress that there are no claims like this neither in section 2 of our manuscript, nor in any other section. Moreover, in section 2, we do not present any effective theory. Section 2 is a very short review of the physics of magnetic charges, with emphasis on the various theoretical arguments for their existence. We do not claim any new results in this section, except for the very last sentences of it where we do outline some outcomes from the subsequent sections, but do not write anything about effective theories of any kind. Later in the article, we present the effective field theory (EFT) of the interactions between axions and electromagnetic fields, however we do not claim that "we cannot write the effective theory for the monopole", contrary to the assertion by the referee. Therefore, the referee criticizes the point which is not present in our manuscript, and so the critique is irrelevant.

Objection 2

Let us now consider the second paragraph of the point 1 of the referee report. The referee asserts that the EFT of the interactions between axions and electromagnetic fields that we present in section 5 is not under perturbative control due to the large unknown contribution of the virtual processes involving magnetic charges:

The problem is that validity of this effective description is hard to verify, as the theory carrying the elementary particles with both types of charges (electric and magnetic) is expected to be strongly coupled and thereby out of control. For magnetically charged solitons in weakly coupled theory, such as 't Hooft-Polyakov monopoles, this is not an issue due to exponential suppression of their contribution into the virtual processes. But, for elementary particles with magnetic charges, the theory enters the strong coupling regime. In particular, the validity of classical equations of motions (e.g., motion of a point particle) cannot be trusted due to strong quantum back-reaction which is un-calculable. Consequently, the conclusions driven from such an effective description are unreliable.

We disagree with this claim by the referee. This claim would be true if we had considered the theory with very light magnetic monopoles, having masses that are not too much larger than the energy scale probed by the axion search experiments. However, this is absolutely not the case. First of all, obviously, such light magnetic monopoles are ruled out by experiments and observations. Second, the masses of the monopoles we consider are of order of the axion decay constant $f_a$, and from Fig. 1 of our manuscript, one can see that in the phenomenologically interesting region, the masses of these monopoles are many ($\gtrsim 10$) orders of magnitude larger than even the electroweak scale. The loop effects of these monopoles are therefore hugely suppressed by the powers of a very small parameter $gE/M \ll 1$, where $g$ is the magnetic charge, $E$ is the energy scale of axion experiments, and $M$ is the monopole mass. Thus, such loop corrections are absolutely negligible and do not invalidate the EFT that we consider in section 5. This EFT is fully under perturbative control. There is therefore no uncalculable quantum backreaction, contrary to what the referee asserts, and the classical equations of motion we derive in section 5 are valid as an adequate low energy description of the interactions between axions and electromagnetic fields.

2. Analysis of the second issue raised by the referee

The referee starts the discussion of the second point by interpreting our results in the following way:

Next, the authors consider the extra couplings with electromagnetism. From the point of view of UV-completion, such couplings are possible if the axion is shared between two Yang-Mills sectors (the authors acknowledge this on page 28 at the end of the second paragraph).

We strongly disagree with the latter statement. The axion couplings which we consider can well arise in the models which have only one Yang-Mills sector, and in fact there is no need at all to introduce additional Yang-Mills sectors for the results of our work to hold. In the manuscript, we do not discuss models with additional Yang-Mills sectors. The referee mentions the end of the second paragraph on page 28 of our manuscript to support his claim. However, in this paragraph, we explicitly state what kind of gauge theories we consider, in particular we write: "Let us then consider the case where the interaction Lagrangian (5.9) describes the Higgs phase of a non-Abelian gauge theory". Since the interaction Lagrangian (5.9) contains the electromagnetic field, the simplest case of such non-Abelian theory would be a Grand Unified Theory (GUT) based on a simple gauge group, and therefore having only one Yang-Mills sector. Thus, the referee criticizes a point which is not present in our manuscript, and so the critique is irrelevant.

Finally, although we have just showed that the second issue raised by the referee has no relevance for our work, let us indicate two further inaccuracies within the corresponding argument by the referee, in order to avoid any possible misunderstanding of our results. First, the second paragraph on page 28, mentioned by the referee, as well as the whole subsection 5.2, discusses only one particular kind of coupling between axions and electromagnetic field -- the Witten-effect induced coupling -- and so it is incorrect to assert that the statements from this subsection hold for any other couplings as well. Second, even if one did consider a model with multiple Yang-Mills sectors, it is not true that the solution to the strong CP problem would necessarily be jeopardized. Extra contribution to the axion potential need not shift the minimum of the potential. If the minimum is not shifted, the Peccei-Quinn solution to the strong CP problem still works perfectly.

To conclude, we showed that the issues raised by the referee have no relevance for our work and certainly do not invalidate the results we obtained. In particular, the first of the three objections by the referee is raised against the statement we never made in the manuscript in the first place. The second objection is irrelevant as long as one considers magnetic charges that are sufficiently heavy, which is the only realistic case given the experimental data we have. Finally, the third objection stems from an incorrect, too restrictive, interpretation of our analysis of the possible origin of the Witten-effect induced axion coupling, moreover the third objection is based on an incorrect argument.

Let us also draw attention to the fact that none of these would-be fundamental inconsistencies at a conceptual level have been indicated by Prof. Reece in the other referee report on this manuscript.

---

## Editorial Decision

rejected_or_withdrawn